# S2AP: SCORE-SPACE SHARPNESS MINIMIZATION FOR ADVERSARIAL PRUNING

## ABSTRACT

Adversarial pruning methods have emerged as a powerful tool for compressing neural networks while preserving robustness against adversarial attacks. These methods typically follow a three-step pipeline: (i) pretrain a robust model, (ii) select a binary mask for weight pruning, and (iii) finetune the pruned model. To select the binary mask, these methods minimize a robust loss by assigning an importance score to each weight, and then keep the weights with the highest scores. However, this score-space optimization can lead to sharp local minima in the robust loss landscape and, in turn, to an unstable mask selection, reducing the robustness of adversarial pruning methods. To overcome this issue, we propose a novel plug-in method for adversarial pruning, termed Score-space Sharpness-aware Adversarial Pruning (S2AP). Through our method, we introduce the concept of score-space sharpness minimization, which operates during the mask search by perturbing importance scores and minimizing the corresponding robust loss. Extensive experiments across various datasets, models, and sparsity levels demonstrate that S2AP effectively minimizes sharpness in score space, stabilizing the mask selection, and ultimately improving the robustness of adversarial pruning methods.

## 1 INTRODUCTION

Deep neural networks are susceptible to adversarial attacks, which entail optimizing an input perturbation added to the original sample to induce a misclassification (Biggio et al., 2013; Szegedy et al., 2014). Besides robustness against adversarial examples, networks are often required to be compact and suitable for resource-constrained scenarios (Liu & Wang, 2023), where the model's dimension cannot be chosen at hand but requires respecting a given constraint. In this regard, neural network pruning (LeCun et al., 1989) represents a powerful compression method by removing redundant or less impactful parameters according to a desired sparsity rate and, as a result, allowing the preservation of much of the performance of a dense model counterpart (Blalock et al., 2020).

Adversarial Pruning (AP) methods aim to fulfill this twofold requirement, thus extending model compression to the adversarial case, by removing parameters less responsible for adversarial robustness drops (Piras et al., 2024). While prior work extended naïve pruning heuristics to robustness, such as based on the lowest weight magnitude (LWM) of robust models (Han et al., 2015; Sehwag et al., 2019), recent approaches proposed different strategies to quantify each parameter's importance, and thus select an optimized pruning mask accordingly. These methods, such as HARP (Zhao & Wressnegger, 2023) and HYDRA (Sehwag et al., 2020), use real-valued importance scores, one for each model's weight, indicating how much robust loss degrades based on that parameter's removal. These scores are then optimized during the pruning stage by: (i) computing the robust loss using the top-$k$ parameters in the forward pass (where $k$ is the desired sparsity); and (ii) updating each parameter's importance in the backward pass. This procedure circumvents the intractability of the binary mask optimization problem imposed by the $\ell_0$ constraint on the weights (i.e., the desired sparsity). Hence, it enables a parameter selection process based on the scores minimizing a robust objective, yielding a final mask with enhanced adversarial robustness. However, the final subnet is determined by a *discrete* top-$k$ operator applied to these *continuous* scores. Consequently, the effectiveness of the pruning mask in preserving robustness is strongly dependent on importance-score optimization. Small score variations near the pruning threshold can swap the ordering of scores and flip many entries of the binary mask, leading to significant changes in the selected top-$k$ parameters and volatile robustness. This sensitivity highlights the need for a smoother, more stable *score-space*

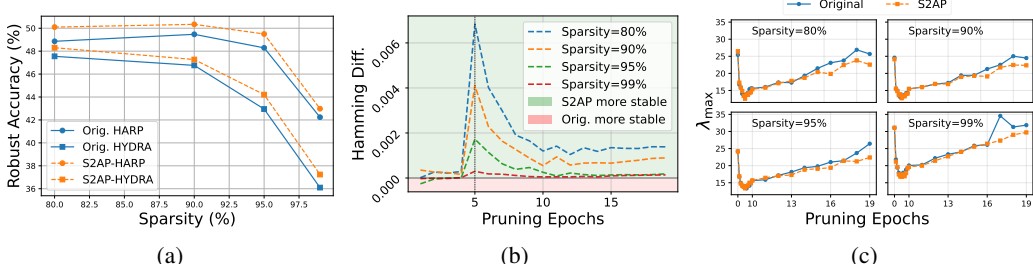

(a)  (b)  (c)

Figure 1: Effects of S2AP on a ResNet18 CIFAR10 model. (a) Improved robust accuracy of pruned models. (b) Enhanced mask stability (quantified as Hamming distance $h$, i.e., measuring how much each mask $\boldsymbol{m}_t$ across pruning epochs changes compared to the first computed mask $\boldsymbol{m}_0$). We subtract and plot $h_{orig} - h_{S2AP}$, thus yielding positive values where S2AP is more stable (green area), and negative values vice versa (red area). S2AP enhances mask stability, particularly after pruning epoch 5 when warm-up ends and explicit sharpness minimization begins. (c) Minimized sharpness in the robust loss landscape (where the largest eigenvalue $\lambda_{max}$ indicates more sharpness).

optimization landscape. In this regard, recent advances in neural network training suggest that explicitly minimizing sharpness in the loss landscape can foster not only generalization (Foret et al., 2021), but also adversarial robustness (Wu et al., 2020; Stutz et al., 2021). These approaches, such as Adversarial Weight Perturbations (AWP) (Wu et al., 2020) for adversarial robustness, work by perturbing the network parameters (i.e., the weights) and minimizing the corresponding loss to reduce sharpness and improve performance.

Inspired by these findings, we extend the concept of sharpness minimization in adversarial robustness beyond the traditional parameter-space setting, where weights are perturbed, to the novel context of importance score optimization. We thereby propose a *score-space* sharpness minimization approach for adversarial pruning methods, that we define as Score-space Sharpness-aware Adversarial Pruning (S2AP), which reduces the sharpness of the loss landscape parameterized by importance scores, stabilizing the mask selection and improving adversarial robustness of pruned models. Crucially, S2AP is implemented as a plug-in, allowing seamless integration into existing AP methods (and any other score-based approach) without altering their core logic or loss formulation. Overall, our main contributions are organized as follows:

(i) we present the S2AP method (Sect. 3), discussing its algorithm in a step-by-step approach;

(ii) we then demonstrate, across multiple architectures, datasets, and sparsity rates, how S2AP improves robustness of adversarial pruning methods (Sect. 4.2);

(iii) we finally show, on the same comprehensive setup, the minimized sharpness in the score-space landscape and the induced mask search stability (Sect. 4.3).

A preview of the discussed effects and results can be seen in Figure 1, where we show the improved robustness of S2AP (Figure 1a), the stabilized mask selection based on the masks' Hamming distances (Figure 1b), and the minimized sharpness based on the largest eigenvalue (Figure 1c).

## 2 ADVERSARIAL PRUNING AND SCORE-SPACE

**Notation.** Let us denote with $\mathcal{D} = \{(\boldsymbol{x}_i, y_i)\}_{i=1}^{n}$ a training set of $n$ $d$-dimensional samples $\boldsymbol{x} \in \mathcal{X} = [0,1]^d$ along with their labels $y \in \mathcal{Y} = \{1, \ldots, C\}$. For a network $f(\cdot\,;\boldsymbol{w})$ with parameters $\boldsymbol{w} \in \mathbb{R}^p$, we define the average loss function computed on the dataset $\mathcal{D}$ (or on a batch) as $\mathcal{L}(\boldsymbol{w}, \mathcal{D}) = 1/n \sum_i \ell(y_i, f(\boldsymbol{x}_i; \boldsymbol{w}))$, being $\ell$ any suitable sample-wise loss and $f$ the $C$ logits of the network.

**Adversarial Robustness.** Machine Learning (ML) models are susceptible to adversarial attacks (Biggio et al., 2013; Szegedy et al., 2014), which create input samples misclassified by the attacked model. In this regard, adversarial training is considered the go-to defense, minimizing a

given robust loss $\hat{\mathcal{L}}$ defined as the inner maximization in the following optimization problem:

$$\min_{\boldsymbol{w}} \ \hat{\mathcal{L}}(\boldsymbol{w}, \mathcal{D}), \quad \hat{\mathcal{L}}(\boldsymbol{w}, \mathcal{D}) = \frac{1}{n} \sum_{i=1}^{n} \max_{\|\boldsymbol{\delta}_i\| \leq \epsilon} \ell(y_i, f(\boldsymbol{x}_i + \boldsymbol{\delta}_i; \boldsymbol{w})), \quad (1)$$

where $\boldsymbol{x}_i + \boldsymbol{\delta}_i \in [0,1]^d, \forall i$, i.e., that each perturbed sample still lies in the sample space upon adding an adversarial perturbation $\boldsymbol{\delta}$ bounded by a given $\ell_p$ bound $\epsilon$.

**Adversarial Pruning Methods.** Pruning aims to reduce the size of a network by removing its parameters (e.g., weights) while preserving performance (LeCun et al., 1989). Similarly, Adversarial Pruning (AP) methods aim to reduce model size while preserving robustness against adversarial attacks (Piras et al., 2024). Recent AP methods proposed solving the following optimization problem:

$$\min_{\|\boldsymbol{m}\|_0 \leq k} \hat{\mathcal{L}}(\boldsymbol{w} \odot \boldsymbol{m}, \mathcal{D}), \quad (2)$$

where $\boldsymbol{m} \in \{0,1\}^p$ is a $p$-dimensional mask constrained to have $k$ non-zero entries. The mask is element-wise multiplied by the weights $\boldsymbol{w}$, ensuring that the pruned model satisfies the sparsity rate $k$. However, the sparsity constraint makes Eq. 2 a non-convex, combinatorial problem. AP methods like HARP (Zhao & Wressnegger, 2023), HYDRA (Sehwag et al., 2020), thus solve it by relaxing the sparsity constraint through the use of *importance scores*.

**Importance Scores.** During the pruning stage, while weights are kept invariant, optimizing importance scores amounts to defining a vector of continuous values $\boldsymbol{s} \in \mathbb{R}^p$, initialized proportionally to the weights, which are learnable and optimized with respect to the robust loss $\hat{\mathcal{L}}$ as follows:

$$\min_{\boldsymbol{s}} \ \hat{\mathcal{L}}(\boldsymbol{w} \odot M(\boldsymbol{s}, k), \mathcal{D}), \quad (3)$$

where $\hat{\mathcal{L}}$ is computed, given $k$, through a masking function $M(\boldsymbol{s}, k)$ that assigns 1 only to the top-k entries of $\boldsymbol{s}$, thus imposing sparsity. Let us remark that such an optimization procedure is non-trivial: in the forward pass, the loss is computed using the top-$k$ parameters as $\hat{\mathcal{L}}(\boldsymbol{w} \odot M(\boldsymbol{s}, k), \mathcal{D})$; during backpropagation, these methods adopt a straight-through estimator (STE) substituting $\partial M/\partial \boldsymbol{s}$ with 1 (i.e., the identity) following Ramanujan et al. (2020). This method enables propagating the gradient through the non-differentiable mask and optimizing each score according to its importance. We thus define as **score-space** the $p$-dimensional space $\mathbb{R}^p$ spanned by the importance scores $\boldsymbol{s}$, and study the robust loss landscape $\hat{\mathcal{L}}(\boldsymbol{w} \odot M(\boldsymbol{s}, k), \mathcal{D})$ defined over it.

**Formulation Generality.** The formulation of Eq. 3 encompasses all AP methods based on importance-score optimization. Different methods can, however, define different loss functions (that we generalize through $\hat{\mathcal{L}}$). This is the case of HARP (Zhao & Wressnegger, 2023), which defines additional penalty terms allowing for optimizing layer-wise sparsity. We specify that our formulation unifies different loss formulations from various AP methods, and as we will describe in the next section, the proposed S2AP can "wrap" any AP method based on importance-score optimization.

## 3 S2AP: Minimizing Score-Space Sharpness

From Sect. 2, it becomes evident that score optimization on a robust loss is the core logic of adversarial pruning. We improve such an approach by minimizing score-space sharpness. Hence, our Score-space Sharpness-aware Adversarial Pruning (S2AP) method avoids that small score shifts induce relevant mask changes, thus stabilizing the pruning process and increasing adversarial robustness. Following Eq. 3, and given the sharpness minimization approach from Wu et al. (2020), we define the S2AP problem as follows:

$$\boldsymbol{s}^* \in \arg\min_{\boldsymbol{s}} \max_{\boldsymbol{z}} \hat{\mathcal{L}}(\boldsymbol{w} \odot M(\boldsymbol{s} + \boldsymbol{z}, k), \mathcal{D}), \quad (4)$$

$$\text{where } \|\boldsymbol{z}_l\| \leq \gamma \|\boldsymbol{s}_l\|, \quad (5)$$

and $\gamma$ constraints the *score perturbation $\boldsymbol{z}$* applied on $\boldsymbol{s}$, scaling it w.r.t. the norm of the scores of each layer $l$. S2AP solves such optimization through Algorithm 1, as detailed below. Note that the sections of Algorithm 1 outside the orange box are common to AP methods (cf. Sect. 2).

**Algorithm 1:** Score-Sharpness-aware Adversarial Pruning.

---

**Input** : $\boldsymbol{w} \in \mathbb{R}^p$, initial weights; $\boldsymbol{s} \in \mathbb{R}^p$, set of importance scores; $M(\boldsymbol{s}, k)$, masking function with pruning rate $k$; $\boldsymbol{x}$, training inputs samples; $\eta$, learning rate; $I$, number of iterations; $L$, number of layers; $\gamma$, perturbation scaling factor; $\hat{\mathcal{L}}$, robust loss.

**Output:** Binary mask $\boldsymbol{m}^* \in \{0,1\}^d$.

1 Initialize parameters $\boldsymbol{s} = \texttt{scale}(\boldsymbol{w})$, $\boldsymbol{x}_i' \leftarrow \boldsymbol{x}$, $\boldsymbol{s}^* \leftarrow \boldsymbol{s}$, $\boldsymbol{z} \leftarrow 0$

2 **for** $i \leftarrow 1$ **to** $I$ **do**

3      Generate adversarial examples on pruned model $\boldsymbol{x}_i' \leftarrow \boldsymbol{x}_i + \boldsymbol{\delta}_i$

4      Compute robust loss on pruned model $\hat{\mathcal{L}}(\boldsymbol{s}) = \hat{\mathcal{L}}(\boldsymbol{w} \odot M(\boldsymbol{s}, k), \mathcal{D})$

5

6      Generate score-space perturbation $\boldsymbol{z} \leftarrow \boldsymbol{z} + \eta(\nabla_z \hat{\mathcal{L}}(\boldsymbol{s} + \boldsymbol{z})/\|\nabla_z \hat{\mathcal{L}}(\boldsymbol{s} + \boldsymbol{z})\|)$

7      **for** $l \leftarrow 1$ **to** $L$ **do**

8          **if** $\|\boldsymbol{z}^{(l)}\| > \gamma \|\boldsymbol{s}^{(l)}\|$ **then**

9             Project perturbation $\boldsymbol{z}^{(l)} \leftarrow \left(\gamma \|\boldsymbol{s}^{(l)}\| / \|\boldsymbol{z}^{(l)}\|\right) \boldsymbol{z}^{(l)}$

10      Update scores $\boldsymbol{s} \leftarrow \boldsymbol{s} - \eta(\nabla_s \hat{\mathcal{L}}(\boldsymbol{s} + \boldsymbol{z})/\|\nabla_s \hat{\mathcal{L}}(\boldsymbol{s} + \boldsymbol{z})\|)$

11      Restore scores $\boldsymbol{s} \leftarrow \boldsymbol{s} - \boldsymbol{z}$         **S2AP**

12

13      **if** $\hat{\mathcal{L}}(\boldsymbol{s}) < \hat{\mathcal{L}}(\boldsymbol{s}^*)$ **then**

14          Update best loss $\hat{\mathcal{L}}(\boldsymbol{s}^*) \leftarrow \hat{\mathcal{L}}(\boldsymbol{s})$

15 **return** $\boldsymbol{m}^* \leftarrow M(\boldsymbol{s}^*, k)$

---

**Generating Adversarial Examples.** We initialize, in line 1, the set of importance scores $\boldsymbol{s}$ proportionally to $\boldsymbol{w}$ through scale, which scales the scores proportionally to the weights' magnitude. This enables creating a pruned model ($f(\boldsymbol{w} \odot M(\boldsymbol{s}, k))$) through which we compute the adversarial examples $\boldsymbol{x}'$ (line 3) using the $\ell_\infty$ PGD attack (Madry et al., 2018). Following Eq. 1, we thus craft a perturbation $\boldsymbol{\delta}$ constrained on $\epsilon$. Computing the adversarial examples allows defining a robust loss $\hat{\mathcal{L}}$ which we denote, for brevity and emphasis on the scores, as $\hat{\mathcal{L}}(\boldsymbol{s})$ in line 4.

**Score-Space Perturbation.** Defining a robust loss and creating adversarial examples is a common step of score-based AP methods. During the pruning stage, in fact, these methods' weights are left unchanged while importance scores $\boldsymbol{s}$ are optimized according to a robust objective to find the best mask $\boldsymbol{m} = M(\boldsymbol{s}, k)$. Through S2AP, we are interested in minimizing the score-space sharpness. Hence, before the standard score optimization, when using S2AP we craft a score-space perturbation (line 6) in one single iteration, aiming to shift the loss in score space from the $i$-th iteration's local minima towards a point of higher loss. We thus create a *worst-case* score perturbation.

In line 9, we iterate over the $L$ layers of the network and project our perturbation $\boldsymbol{z}$ in a bound defined by $\gamma$. More precisely, according to the layer's score magnitude $\|\boldsymbol{s}^{(l)}\|$, we scale $\boldsymbol{z}^{(l)}$ to $(\gamma \|\boldsymbol{s}^{(l)}\| / \|\boldsymbol{z}^{(l)}\|) \boldsymbol{z}^{(l)}$ if $\|\boldsymbol{z}^{(l)}\| > \gamma \|\boldsymbol{s}^{(l)}\|$, which corresponds to projecting back the perturbation into the "ball" defined by $\gamma$ when exceeding, and leave as is otherwise. The layer-wise projection primarily addresses the numeric differences across layers. Without per-layer scaling, the magnitude of the generated perturbation $\boldsymbol{z}$ can be perceived differently across layers, leading to either no effect or numerical overflow. A layer-wise projection instead keeps every layer's perturbation proportional to its current score norm, preserving well-conditioned updates and preventing disparity across layers.

**Score Update.** Once the score perturbation $\boldsymbol{z}$ is computed, we evaluate the gradient at the perturbed scores $\boldsymbol{s} + \boldsymbol{z}$ (line 10), and take an optimization step to move $\boldsymbol{s}$ in the direction that, in turn, reduces sharpness. After optimizing $\hat{\mathcal{L}}$, S2AP ends by removing the previously applied perturbation to restore the original reference point $\boldsymbol{s}$ for the next iteration (line 11). We specify that also the score update of line 10 is common to AP methods. However, instead of updating scores based on the loss computed on score space $\hat{\mathcal{L}}(\boldsymbol{s})$, S2AP enables a "sharpness-aware" update on perturbed score space $\hat{\mathcal{L}}(\boldsymbol{s} + \boldsymbol{z})$. Finally, through line 14 and line 15, we save $\boldsymbol{s}^*$ corresponding to the lowest $\hat{\mathcal{L}}$ and return the best mask $\boldsymbol{m}^*$ via the function $M(\boldsymbol{s}^*, k)$, which is finally multiplied to the pretrained weights.

**S2AP Finetuning.** After defining mask $\boldsymbol{m}^*$ and pruning the model, some of the AP methods we enhance with S2AP finetune the pruned weights to restore performance using a robust objective (Han et al., 2015). In S2AP, we choose to finetune the pruned model by aligning the objective with the score-space sharpness minimization implemented while pruning. Hence, we choose to minimize sharpness using the AWP (Wu et al., 2020) approach applied on the classical weight-space:

$$\boldsymbol{w}^* \in \arg\min_{\boldsymbol{w}} \max_{\boldsymbol{\nu}} \hat{\mathcal{L}}((\boldsymbol{w} + \boldsymbol{\nu}) \odot \boldsymbol{m}^*), \tag{6}$$

$$\text{where } \|\boldsymbol{\nu}_l\| \leq \gamma \|\boldsymbol{w}_l\|, \tag{7}$$

and $\nu$, in this case, is a weight perturbation added to the preserved weights according to $\boldsymbol{m}^*$ found through Algorithm 1. Therefore, instead of perturbing all the weights as in typical sharpness minimization, we add a perturbation only to the top-$k$ weights according to the mask found in the previous step, and project the perturbation based on the layers' weight magnitude. We provide a more details in Sect. A.2, and show S2AP's performance independence in Table 5.

## 4 EXPERIMENTS

S2AP minimizes score-space sharpness, building upon the observation that a smoother loss landscape enhances adversarial robustness. In turn, after describing the general experimental setup (Sect. 4.1), we show and discuss the robustness of S2AP on adversarial pruning methods (Sect. 4.2), and then analyze the effect of S2AP on score-space sharpness minimization and mask selection stability (Sect. 4.3). More experiments can be found in Appendix A, Appendix B, and Appendix C.

### 4.1 EXPERIMENTAL SETUP

**AP Methods, Models, and Datasets.** We test S2AP on the HARP, HYDRA, and Robust-Lottery Ticket Hypothesis (RLTH) adversarial pruning methods (Zhao & Wressnegger, 2023; Sehwag et al., 2020; Fu et al., 2021), while comparing to the original implementations (Orig.). These approaches are all based on the optimization of importance scores summarized in Eq. 3. However, while HARP and HYDRA start from a robust pretrained model, and, after pruning, finetune the pruned model, RLTH tests the LTH on a randomly initialized model and does not finetune the resulting pruned parameterization. We show RLTH results in Appendix B. We choose $80\%, 90\%, 95\%$, and $99\%$ as sparsity rates, indicating the rate of pruned parameters. We employ the ResNet18 (He et al., 2016), VGG16 (Simonyan & Zisserman, 2015), and WideResNet-28-4 (Zagoruyko & Komodakis, 2016) architectures on both the CIFAR10 (Krizhevsky et al., 2009) and SVHN (Netzer et al., 2011) datasets. In addition, we test HARP and HYDRA on the larger-scale ImageNet (Deng et al., 2009) dataset using the ResNet50 architecture (we refrain from testing RLTH on ImageNet, as with no finetuning, the accuracy is too low with moderate epochs). Finally, we prune a vision transformer (ViT) with a patch size of $4 \times 4$, resulting in 64 tokens for $32 \times 32$ images, to $20\%, 40\%$, and $60\%$ sparsity. It comprises 8 transformer layers, 6 attention heads per layer, and a hidden dimensionality of 384. The MLP blocks have an expansion ratio of 4, with a hidden dimension of 1536.

**Adversarial Training and Evaluation.** We pretrain, prune, and finetune the models with HARP and HYDRA (prune only for RLTH) using the TRADES loss (Zhang et al., 2019) (pretrained models' results are shown in Sect. A.1). During adversarial training, we generate adversarial examples using $\ell_\infty$ PGD-10 with perturbation size $\epsilon = 8/255$ and step-size $\alpha = 2/255$. Similarly, we evaluate robustness using the AutoAttack (AA (Croce & Hein, 2020)) ensemble with $\ell_\infty$ perturbation bound $\epsilon = 8/255$ for every adversarial robustness evaluation. For HARP and HYDRA, we pretrain and finetune in 100 epochs, while we prune for 20 epochs. Also, we search for the RLTH tickets in 20 epochs. Of these 20 epochs, for each method, S2AP is applied after 5 warm-up epochs. For completeness, we discuss the computational cost of pruning with S2AP in Sect. A.4.

**S2AP Setup.** We use the same adversarial training setup as the original methods to prune with S2AP. Also, we find one step to be sufficient for finding a score perturbation, as in Wu et al. (2020). However, we must specify a $\gamma$ value to design the layer-wise perturbation projection. For ResNet18 and WideResNet on CIFAR10, we set $\gamma = 0.001$; for VGG16 on CIFAR10 and SVHN, $\gamma = 0.0025$; for ResNet18 on SVHN, $\gamma = 0.0075$; for WideResNet on SVHN, $\gamma = 0.005$; and finally, for ResNet50 on ImageNet, we set $\gamma = 0.0075$. The same $\gamma$ is used to bound weight perturbation for S2AP finetuning in HARP and HYDRA. For ViTs, we choose gamma 0.0025. We select the $\gamma$ value according to the highest robust accuracy, and discuss its selection in Sect. A.3.

Table 1: CIFAR-10 results. We show the clean/robust$_{\pm std}$ accuracy and the difference between Orig. and S2AP robust generalization gap ($\Delta$). In bold, the model with the highest robustness.

| Network | Sparsity | HARP | | | HYDRA | | |
|---|---|---|---|---|---|---|---|
| | | Orig. | S2AP | Gap $\Delta$ | Orig. | S2AP | Gap $\Delta$ |
| ResNet18 | 80% | 81.26 / 48.86$_{\pm 0.16}$ | **81.36 / 50.10**$_{\pm 0.21}$ | +1.14 | 80.73 / 47.55$_{\pm 0.81}$ | **81.47 / 48.30**$_{\pm 0.91}$ | +0.01 |
| | 90% | 81.62 / 49.47$_{\pm 0.24}$ | **82.10 / 50.34**$_{\pm 0.33}$ | +0.39 | 80.85 / 46.76$_{\pm 1.34}$ | **80.89 / 47.27**$_{\pm 1.09}$ | +0.47 |
| | 95% | 82.88 / 48.29$_{\pm 0.44}$ | **82.68 / 49.50**$_{\pm 0.46}$ | +1.41 | 80.83 / 42.95$_{\pm 1.38}$ | **80.14 / 44.21**$_{\pm 0.92}$ | +1.95 |
| | 99% | 80.72 / 42.24$_{\pm 0.13}$ | **81.46 / 42.98**$_{\pm 0.39}$ | +0.00 | 80.51 / 36.10$_{\pm 1.41}$ | **80.93 / 37.24**$_{\pm 1.20}$ | +0.72 |
| VGG16 | 80% | 78.49 / 45.20$_{\pm 0.69}$ | **79.19 / 45.93**$_{\pm 0.34}$ | +0.03 | 77.10 / 44.63$_{\pm 0.09}$ | **78.70 / 44.95**$_{\pm 0.12}$ | -1.28 |
| | 90% | 80.54 / 45.53$_{\pm 0.47}$ | **78.64 / 46.26**$_{\pm 0.41}$ | +2.63 | 77.65 / 43.07$_{\pm 0.23}$ | **77.07 / 43.57**$_{\pm 0.06}$ | +1.08 |
| | 95% | 78.70 / 44.74$_{\pm 0.23}$ | **79.12 / 45.67**$_{\pm 0.11}$ | +0.51 | 76.79 / 40.75$_{\pm 0.23}$ | **76.55 / 41.48**$_{\pm 0.83}$ | +0.97 |
| | 99% | 77.85 / 41.38$_{\pm 0.88}$ | **78.61 / 42.04**$_{\pm 0.36}$ | -0.10 | 75.10 / 33.24$_{\pm 1.44}$ | **76.43 / 34.09**$_{\pm 1.04}$ | -0.48 |
| WRN28-4 | 80% | 81.69 / 50.08$_{\pm 0.67}$ | **81.73 / 51.28**$_{\pm 0.74}$ | +1.16 | 81.94 / 50.17$_{\pm 0.68}$ | **82.37 / 50.79**$_{\pm 0.47}$ | +0.19 |
| | 90% | 82.02 / 50.52$_{\pm 0.51}$ | **82.31 / 51.83**$_{\pm 0.71}$ | +1.02 | 81.24 / 50.17$_{\pm 0.35}$ | **82.29 / 50.40**$_{\pm 0.67}$ | -0.82 |
| | 95% | 82.47 / 50.57$_{\pm 0.76}$ | **82.49 / 51.04**$_{\pm 0.58}$ | +0.45 | 81.42 / 49.22$_{\pm 0.21}$ | **81.90 / 49.40**$_{\pm 0.78}$ | -0.30 |
| | 99% | 76.14 / 44.68$_{\pm 0.82}$ | **76.29 / 44.93**$_{\pm 0.27}$ | +0.10 | **74.66 / 42.28**$_{\pm 0.58}$ | 74.00 / 42.01$_{\pm 0.64}$ | +0.39 |

Table 2: SVHN results. We show the clean/robust$_{\pm std}$ accuracy and the difference between Orig. and S2AP robust generalization gap ($\Delta$). In bold, the model with the highest robustness.

| Network | Sparsity | HARP | | | HYDRA | | |
|---|---|---|---|---|---|---|---|
| | | Orig. | S2AP | Gap $\Delta$ | Orig. | S2AP | Gap $\Delta$ |
| ResNet18 | 80% | 92.55 / 40.06$_{\pm 1.03}$ | **91.53 / 41.50**$_{\pm 1.05}$ | +2.46 | 92.71 / 42.56$_{\pm 1.02}$ | **92.69 / 43.72**$_{\pm 1.07}$ | +1.18 |
| | 90% | 91.61 / 40.14$_{\pm 0.82}$ | **91.07 / 41.33**$_{\pm 0.26}$ | +1.73 | **91.90 / 41.83**$_{\pm 0.65}$ | 91.63 / 41.58$_{\pm 0.30}$ | +0.02 |
| | 95% | 87.53 / 38.16$_{\pm 0.66}$ | **88.68 / 38.75**$_{\pm 0.19}$ | -0.56 | 90.33 / 40.53$_{\pm 0.16}$ | **90.63 / 40.86**$_{\pm 0.28}$ | +0.03 |
| | 99% | 88.42 / 35.24$_{\pm 0.57}$ | **89.71 / 36.12**$_{\pm 0.76}$ | -0.41 | 87.89 / 40.83$_{\pm 0.83}$ | **88.63 / 41.10**$_{\pm 0.28}$ | -0.47 |
| VGG16 | 80% | 86.36 / 47.28$_{\pm 1.11}$ | **87.80 / 49.69**$_{\pm 1.05}$ | +0.97 | 85.75 / 46.13$_{\pm 1.19}$ | **87.64 / 48.95**$_{\pm 1.16}$ | +0.93 |
| | 90% | 87.58 / 49.16$_{\pm 1.12}$ | **87.77 / 49.49**$_{\pm 1.19}$ | +0.14 | 86.22 / 48.04$_{\pm 0.81}$ | **87.09 / 48.12**$_{\pm 0.22}$ | -0.79 |
| | 95% | 86.95 / 49.16$_{\pm 0.29}$ | **86.98 / 49.28**$_{\pm 0.58}$ | +0.09 | 86.10 / 45.95$_{\pm 0.83}$ | **85.03 / 47.12**$_{\pm 0.63}$ | +2.24 |
| | 99% | 84.93 / 46.33$_{\pm 0.51}$ | **84.73 / 46.61**$_{\pm 0.27}$ | +0.48 | **83.12 / 41.52**$_{\pm 0.72}$ | 81.59 / 41.39$_{\pm 0.46}$ | +1.40 |
| WRN28-4 | 80% | 90.01 / 36.73$_{\pm 1.02}$ | **90.65 / 43.53**$_{\pm 0.61}$ | +6.16 | 95.24 / 42.95$_{\pm 0.84}$ | **88.54 / 44.64**$_{\pm 1.08}$ | +8.39 |
| | 90% | **95.01 / 34.70**$_{\pm 0.91}$ | 92.17 / 31.00$_{\pm 0.76}$ | -0.86 | 93.35 / 36.29$_{\pm 0.39}$ | **91.71 / 38.32**$_{\pm 1.13}$ | +3.67 |
| | 95% | 92.44 / 31.66$_{\pm 0.77}$ | **94.46 / 33.15**$_{\pm 0.72}$ | -0.53 | **89.55 / 43.99**$_{\pm 0.65}$ | 90.43 / 38.89$_{\pm 0.95}$ | -5.98 |
| | 99% | 87.09 / 30.09$_{\pm 0.83}$ | **88.47 / 36.26**$_{\pm 1.12}$ | +4.79 | 93.05 / 31.24$_{\pm 0.49}$ | **85.80 / 42.43**$_{\pm 1.11}$ | +18.44 |

## 4.2 EFFECT OF S2AP ON ADVERSARIAL ROBUSTNESS

S2AP improves the robustness of adversarial pruning methods. We demonstrate such a result through Table 1 for CIFAR10, Table 2 for SVHN, Table 3 for transformers, and finally Table 4 for ImageNet. We further show results using channel pruning in Sect. B.2, and RLTH method in Table 8

**Experimental Results.** In Table 1 for CIFAR10, across every sparsity level and method, S2AP consistently exceeds the robust accuracy of original methods. In general, across models, S2AP improves robustness up to 2 percentage points (p.p.). Importantly, these gains come with improved or negligible drops ($< 0.3$ p.p.) in clean accuracy, as well as low error bars. To provide transparency on the trade-off between clean and robust performance, we also report the clean–robust generalization gap ($\Delta$), defined as the gap of Orig. minus that of S2AP. The

Table 3: ViT on CIFAR-10 and HYDRA: clean / robust accuracy (%) under different sparsity levels. Bold indicates the best between Orig. and S2AP.

| Network | Sparsity (%) | Orig. | S2AP |
|---|---|---|---|
| ViT | 20 | 63.93 / 26.45 | **64.53 / 27.85** |
| | 40 | 63.89 / 25.27 | **64.08 / 26.32** |
| | 50 | 63.02 / 24.71 | **63.87 / 25.86** |

gap measures the relative consistency between clean and robust accuracy, offering insight into how robust performance changes in relation to improvements or drops in clean accuracy. Across all settings, $\Delta$ remains mainly positive, showing that S2AP improves over Orig. without introducing a significant trade-off in generalization. Overall, through our diverse experimental setup, we see the WideResNet model reaching higher robustness compared to the ResNet18 and VGG16 models, but still S2AP consistently outperforming competing methods. A similar conclusion can be drawn for SVHN results in Table 2 and ImageNet results on Table 4. Again, S2AP consistently improves robustness across models, sparsities, and AP methods, with a comparable and often superior stan-

dard accuracy. We extend the S2AP evaluation to Vision Transformers in Table 3. We remark how prior work on adversarial pruning has been limited to standard deep networks, thus marking this as a first experiment of AP methods on transformer-based architecture. We choose to prune with HYDRA, as the HARP method involves optimizing a layer-wise sparsity rate, which is not directly suited for transformer architectures and requires re-thinking the entire method. We prune all linear layers except for the final classification head, which is kept dense to ensure stable output mapping to class logits. We confirm the improved adversarial robustness on such kinds of architectures. Finally, we further validate the efficacy of S2AP by showing results for standard classification accuracy in Sect. B.3, and for robustness against common corruptions in Sect. B.4, thus validating S2AP in more general and external domains.

**Finetuning Ablation Study.** In HARP and HYDRA, after selecting the mask through S2AP, we align the finetuning objective with the pruning one, thus finetuning by perturbing the weights via AWP (Wu et al., 2020), since scores are not used after pruning. We show in Table 5 the "raw" mask adversarial robustness obtained before finetuning, thus the pruned model derived from multiplying the pretrained weights with the mask obtained in Algorithm 1. This comparison enables ablating the finetuning objective and **verifying if the adversarial robustness improvement produced by S2AP is independent from finetuning.** Our results highlight the higher robust accuracy of S2AP against the original

Table 4: ImageNet results using ResNet50 across sparsity levels. Each cell shows clean/robust accuracy.

| Network | Sparsity | Orig. | S2AP |
|---|---|---|---|
| **HARP** | | | |
| ResNet50 | 80% | $61.48 / 33.01_{\pm 0.41}$ | $\mathbf{62.42 / 34.60}_{\pm 0.82}$ |
| | 90% | $54.93 / 24.05_{\pm 0.66}$ | $\mathbf{55.00 / 25.61}_{\pm 0.57}$ |
| | 95% | $40.74 / 21.12_{\pm 0.26}$ | $\mathbf{43.85 / 22.07}_{\pm 0.26}$ |
| | 99% | $28.65 / 12.92_{\pm 0.39}$ | $\mathbf{34.18 / 15.75}_{\pm 0.76}$ |
| **HYDRA** | | | |
| ResNet50 | 80% | $51.36 / 29.71_{\pm 0.48}$ | $\mathbf{56.16 / 31.11}_{\pm 0.39}$ |
| | 90% | $48.11 / 20.13_{\pm 0.33}$ | $\mathbf{54.92 / 24.23}_{\pm 1.17}$ |
| | 95% | $33.29 / 16.43_{\pm 0.67}$ | $\mathbf{34.19 / 17.93}_{\pm 0.82}$ |
| | 99% | $26.07 / 11.40_{\pm 0.20}$ | $\mathbf{27.47 / 12.67}_{\pm 0.59}$ |

AP methods throughout the different network and dataset combinations. In addition, we also discuss minimizing sharpness on the weights' loss landscape, and compare to S2AP, in Sect. B.5.

Table 5: Mask robust accuracy (mean$_{\pm \text{std}}$) on CIFAR10 and SVHN across sparsity levels using ResNet18, VGG-16, and WideResNet-28-4.

| Network | Sparsity | CIFAR10 | | | | SVHN | | | |
|---|---|---|---|---|---|---|---|---|---|
| | | HARP | | HYDRA | | HARP | | HYDRA | |
| | | Orig. | S2AP | Orig. | S2AP | Orig. | S2AP | Orig. | S2AP |
| ResNet18 | 80% | $48.88_{\pm 0.73}$ | $\mathbf{49.55}_{\pm 0.69}$ | $48.56_{\pm 0.66}$ | $\mathbf{48.98}_{\pm 0.75}$ | $46.56_{\pm 0.66}$ | $\mathbf{49.18}_{\pm 0.77}$ | $45.74_{\pm 0.73}$ | $\mathbf{46.11}_{\pm 0.69}$ |
| | 90% | $49.42_{\pm 0.72}$ | $\mathbf{49.60}_{\pm 0.74}$ | $47.41_{\pm 0.84}$ | $\mathbf{48.06}_{\pm 0.71}$ | $\mathbf{49.04}_{\pm 0.79}$ | $48.28_{\pm 0.81}$ | $45.61_{\pm 0.70}$ | $\mathbf{47.62}_{\pm 0.83}$ |
| | 95% | $\mathbf{49.04}_{\pm 0.76}$ | $48.43_{\pm 0.78}$ | $45.55_{\pm 0.91}$ | $\mathbf{45.61}_{\pm 0.86}$ | $41.66_{\pm 1.21}$ | $\mathbf{45.58}_{\pm 0.75}$ | $44.53_{\pm 0.84}$ | $\mathbf{45.14}_{\pm 0.72}$ |
| | 99% | $40.99_{\pm 1.34}$ | $\mathbf{41.86}_{\pm 1.19}$ | $35.15_{\pm 1.48}$ | $\mathbf{36.74}_{\pm 1.42}$ | $40.79_{\pm 0.94}$ | $\mathbf{45.77}_{\pm 1.07}$ | $\mathbf{40.85}_{\pm 0.99}$ | $37.93_{\pm 1.22}$ |
| VGG-16 | 80% | $41.93_{\pm 0.82}$ | $\mathbf{42.84}_{\pm 0.85}$ | $40.31_{\pm 0.95}$ | $\mathbf{41.39}_{\pm 0.91}$ | $46.95_{\pm 0.78}$ | $\mathbf{48.93}_{\pm 0.84}$ | $45.78_{\pm 0.89}$ | $\mathbf{46.17}_{\pm 0.75}$ |
| | 90% | $41.69_{\pm 0.86}$ | $\mathbf{42.11}_{\pm 0.87}$ | $38.12_{\pm 1.12}$ | $\mathbf{40.61}_{\pm 0.93}$ | $\mathbf{47.30}_{\pm 0.79}$ | $46.28_{\pm 0.76}$ | $44.22_{\pm 0.81}$ | $\mathbf{46.17}_{\pm 0.88}$ |
| | 95% | $\mathbf{40.21}_{\pm 0.97}$ | $39.13_{\pm 0.99}$ | $31.81_{\pm 1.42}$ | $\mathbf{38.03}_{\pm 1.08}$ | $46.51_{\pm 0.75}$ | $\mathbf{47.96}_{\pm 0.71}$ | $42.43_{\pm 0.84}$ | $\mathbf{43.78}_{\pm 0.73}$ |
| | 99% | $24.22_{\pm 1.52}$ | $\mathbf{36.41}_{\pm 1.21}$ | $20.54_{\pm 1.68}$ | $\mathbf{29.67}_{\pm 1.49}$ | $43.42_{\pm 0.77}$ | $\mathbf{43.91}_{\pm 0.81}$ | $31.06_{\pm 1.34}$ | $\mathbf{32.64}_{\pm 1.41}$ |
| WRN28-4 | 80% | $50.45_{\pm 0.81}$ | $\mathbf{50.59}_{\pm 0.73}$ | $50.31_{\pm 0.78}$ | $\mathbf{50.41}_{\pm 0.76}$ | $43.79_{\pm 0.74}$ | $\mathbf{47.02}_{\pm 0.78}$ | $\mathbf{49.43}_{\pm 0.73}$ | $47.50_{\pm 0.71}$ |
| | 90% | $50.56_{\pm 0.77}$ | $\mathbf{50.79}_{\pm 0.72}$ | $47.75_{\pm 0.88}$ | $\mathbf{49.30}_{\pm 0.80}$ | $45.89_{\pm 0.75}$ | $\mathbf{46.31}_{\pm 0.74}$ | $43.80_{\pm 0.76}$ | $\mathbf{45.66}_{\pm 0.78}$ |
| | 95% | $49.07_{\pm 0.91}$ | $\mathbf{49.37}_{\pm 0.87}$ | $\mathbf{46.97}_{\pm 0.97}$ | $46.85_{\pm 0.93}$ | $41.69_{\pm 0.79}$ | $\mathbf{45.41}_{\pm 0.76}$ | $48.01_{\pm 0.75}$ | $\mathbf{48.35}_{\pm 0.73}$ |
| | 99% | $38.89_{\pm 1.39}$ | $\mathbf{39.89}_{\pm 1.22}$ | $34.57_{\pm 1.47}$ | $\mathbf{36.30}_{\pm 1.34}$ | $\mathbf{43.58}_{\pm 0.78}$ | $40.87_{\pm 0.81}$ | $\mathbf{40.57}_{\pm 0.79}$ | $38.84_{\pm 0.82}$ |

## 4.3 Effect of S2AP on Score-Space Sharpness and Mask Stability

We evaluate here the effect of S2AP on the sharpness of the loss landscape parameterized by the importance scores. In contrast to conventional approaches, we measure score-space sharpness in the robust loss landscape and adapt the measures accordingly. In addition, we introduce the mask stability property to probe the effect of score-space sharpness minimization on mask-search dynamics. We quantify stability via the normalized Hamming distance between the first and subsequent pruning masks and observe that S2AP generally reduces this distance.

**Minimized Score-Space Sharpness.** We measure score-space sharpness relying on (i) the score-space largest eigenvalue $\lambda_{max}$ measure (Jastrzębski et al., 2017); and (ii) a loss-difference measure addressing the scale-invariance problem of Hessian-based measures (Dinh et al., 2017; Kaur et al., 2023). We measure $\lambda_{max}$ on the score space for each iteration and average the values on each epoch

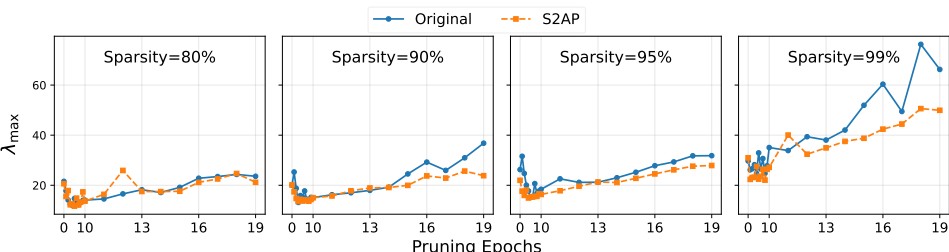

Figure 2: Score-space sharpness measured via largest eigenvalue $\lambda_{max}$ over pruning epochs for HARP on WideResNet28-4 and CIFAR10.

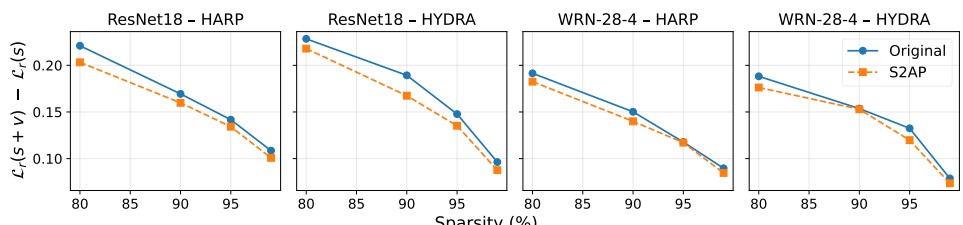

Figure 3: Score-space sharpness measured as difference of perturbed and reference loss values on ResNet18 and WideResNet28-4 CIFAR10 pruned models.

to evaluate sharpness. We show in Figure 1c $\lambda_{max}$ for a ResNet18 model on the CIFAR10 dataset and HARP, which reveals how, across different sparsities, Orig. has the largest eigenvalues (i.e., is sharper) than the S2AP version. The same trend can be validated in Figure 2 for a WideResNet28-4. The loss difference instead is computed by crafting a score-space perturbation added to the scores parameterizing a $\hat{\mathcal{L}}$ minima, and subtracted from the reference $\hat{\mathcal{L}}$ value, thus extending the approach from Andriushchenko et al. (2023); Stutz et al. (2021) to the score space. In this case, we consider the best $\hat{\mathcal{L}}$ minima found during the pruning mask search, then compute the difference $\hat{\mathcal{L}}(s + \boldsymbol{v}) - \hat{\mathcal{L}}(s)$, where $\boldsymbol{v}$ is a score perturbation crafted through the Auto-PGD (APGD) optimization approach. Care must be taken not to conflate this perturbation, added to already optimized scores to simply estimate the loss sharpness, with the one designed in Algorithm 1 added during optimization to induce sharpness. As shown in Figure 3, the sharpness of our S2AP approach is lower. More details on the sharpness measures and additional experiments can be found in Sect. C.1 and Sect. C.2.

**Improved Mask Stability.** Beyond merely flattening the loss landscape, we study a novel property—*mask stability*—to probe the effect of score-space sharpness minimization on mask-search dynamics. We aim to test whether a flatter score-space reduces the sensitivity of the selection to small score-variations (i.e.., whether the mask search becomes less volatile). We capture this phenomenon using the normalized Hamming distance, following prior work that measures mask distances (You et al., 2020). This allows us to compute the differing $0 - 1$ values between binary masks $\boldsymbol{m}$. Hence, over the 20 pruning epochs indexed by $t$, we compute $h = \|\boldsymbol{m}_0 \oplus \boldsymbol{m}_t\|_1 / |\boldsymbol{m}_0|$, where $\oplus$ is a XOR operator measuring the differing bits. For each pruning epoch, we compute $h_{orig} - h_{S2AP}$, and define a positive region, where S2AP is more stable, and a negative region, where the original method is more stable. We show how S2AP improves mask stability for ResNet18 in Figure 1b, while in Figure 4a and Figure 4b we show, respectively, the single Hamming distance curves for original vs. S2AP-based methods and the difference between the curves across all four sparsities. Before the five warm-up epochs, being the overall training procedure identical, numerical differences only result in marginal differences between S2AP and the original methods. Then, the spike registered indicates the immediate increased stability induced by S2AP, which denotes how minimizing sharpness makes the mask selection closer to the first computed mask. As sparsity increases, since a higher sparsity also implies a lower variability of 0's and 1's, the scale of the hamming distance decreases accordingly. More details and additional experiments can be found in Sect. C.3

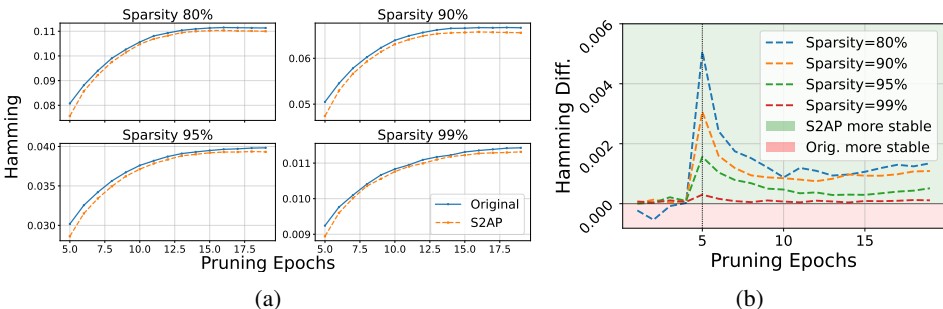

(a)                                       (b)

Figure 4: The Hamming distance for WideResNet28-4 on CIFAR10. In (a) the single hamming distance from epoch 5 of S2AP and Orig. HARP. Lower curves indicate higher stability. In (b), the results from the four (a) subplots by subtracting each Original-S2AP curve, thus yielding a positive-green (negative-red) area where S2AP (Original) methods are more stable.

## 5 RELATED WORK

**Adversarial Robustness and Sharpness.** The work from Wu et al. (2020) first revealed the correlation between robustness and sharpness. In fact, AWP shows that adversarial objectives, such as PGD-AT (Madry et al., 2018), *implicitly* minimize sharpness in the weights' loss landscape. Hence, by *explicitly* minimizing sharpness with respect to both weights and inputs, it improved robustness and flatness. On a larger-scale study by Stutz et al. (2021), and recently also in (Zhang et al., 2024), such a relationship has been investigated in more detail and confirmed thoroughly. In our work, we leverage a similar idea to improve the stability and robustness of adversarial pruning methods.

**Pruning and Sharpness.** Minimizing sharpness through SAM (Foret et al., 2021) has been shown to be beneficial for iterative pruning on BERT models and NLP tasks, compared to the Adam optimizer (Na et al., 2022). The work from Na et al. (2022) has been extended, besides (Lee et al., 2025), to structured pruning and out-of-distribution (OOD) robustness by Bair et al. (2024). The authors prime the network for pruning based on the rationale that a flatter landscape is more prone to pruning. Hence, they develop an adaptive version of SAM by perturbing the channels more likely to be pruned. Further work proposed a single-step sharpness minimization approach aligned with the resource constraints imposed by sparse training (Ji et al., 2024). In contrast, we focus on adversarial robustness (i.e., adversarial pruning) and on score-space sharpness minimization, rather than the typical weights' loss landscape. Most importantly, we do not focus on pre-pruning network priming, but rather explicitly operate on score space during the pruning mask search.

From a conceptual perspective, our work is the first to blend the robustness/sharpness/pruning lines of work by proposing a sharpness minimization approach for adversarial pruning. However, we promote the novel concept of score-space sharpness minimization, thus allowing us to measure and improve mask-search stability, besides robustness.

## 6 CONCLUSIONS, LIMITATIONS, AND FUTURE WORK

We have introduced S2AP, a score-space sharpness minimization for adversarial pruning methods. Leveraging the concept of score-space, S2AP effectively minimizes sharpness, improves the mask-search stability, and consistently increases adversarial robustness across various datasets, models, and sparsities. As limitations, we believe that the additional costs of minimizing sharpness, which apply to all standard SAM-like objectives, might be unsustainable in specific application scenarios. Despite being cost minimization out of this work's scope, we believe "cheaper" approaches such as the one from Ji et al. (2024) could be extended to the S2AP case as future work. Finally, let us specify how the network architecture choices have been dictated by the availability of state-of-the-art AP methods, which do not extend to more recent transformer architectures. Despite being ours, to the best of our knowledge, the first adversarial pruning work considering such architectures, we believe that a consistent setup shift is required for adversarial pruning methods, and hope our work can inspire such improvements. To conclude, we remark how S2AP can be extended to any score-based optimization, beyond adversarial pruning.

**Reproducibility Statement.** We have taken several steps to facilitate reproducibility. The S2AP method is precisely specified in Algorithm 1; the finetuning objective is given Algorithm 2. Our experimental setup—datasets, architectures, sparsity levels, training and evaluation protocols, and threat model—is documented in Sect. 4.1. Hyperparameter choices are reported in the paper and further discussed in Appendix B. We describe the score-space sharpness metrics and the mask-stability metric in Sect. 4.3 with additional implementation details in Appendix C. In the *supplementary material*, we include an anonymized code archive containing all needed source code, training/evaluation scripts, and the *default configurations* used in our experiments; for transparency, these default settings are also listed throughout the paper where relevant and mirrored in the appendix and configuration files. The code will be publicly released upon acceptance.

**Ethics Statement.** We do not identify any ethical concerns associated with this work. Our study does not involve human subjects, user interaction, or personally identifiable information. All experiments use standard, publicly available datasets (CIFAR-10, SVHN, ImageNet) under their respective licenses. The proposed method is defensive—focusing on pruning and adversarial robustness—and does not introduce new attack capabilities beyond standard, widely used evaluation protocols (e.g., PGD, AutoAttack). We are not aware of privacy, security, fairness, or legal compliance issues arising from our methodology or experimental setup, and we have no conflicts of interest or sponsorship to declare. We have read and adhere to the ICLR Code of Ethics.

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

# Supplementary Material for S2AP: Score-space Sharpness Minimization for Adversarial Pruning

The supplementary material is organized as follows:

- **Appendix A**: We discuss additional details for the S2AP method, including pretraining and finetuning details, hyperparameter selection, and overhead computing.

- **Appendix B**: We show additional experiments validating the applicability and effectiveness of S2AP outside the main testbed, including structured pruning, clean standard accuracy, and robustness to corrupted images. We conclude by discussing and showing the comparison of weights and score perturbations during the pruning stage.

- **Appendix C**: We provide additional details and experiments for the eigenvalue computation, the loss difference measuring sharpness, and the mask stability and hamming distance measure.

## A  Additional S2AP Details.

This section describes the additional details concerning our S2AP implementation and results. In detail, we first show the results from the pretrained models used in Table 1, Table 2, Table 4, and Table 5. Then, we discuss in detail the S2AP finetuning algorithm, which concerns perturbing the remaining sparse weight parameterization $\boldsymbol{w} \odot \boldsymbol{m}^*$ as in Eq. 6. We conclude by motivating the choices of the $\gamma$ values bounding the score-perturbations listed in Sect. 4.1, and computing the overhead induced by our S2AP approach compared to a standard score-based pruning optimization.

Let us finally specify that the S2AP **code implementation** is part of the supplementary material and will be publicly released upon paper acceptance.

### A.1  S2AP Pretraining and ImageNet details

We pretrain each CIFAR10 and SVHN model using 100 epochs, and show the resulting adversarial robustness in Table 6. For ImageNet, however, we use the pretrained model provided by Zhao & Wressnegger (2023), and prune for 10 epochs (of which 5 warm-up and 5 S2AP) and finetune for 25 using the Fast Adversarial Training approach.

Table 6: Pretrained models' clean/robust accuracy.

| Model | Dataset | Orig. |
|---|---|---|
| ResNet18 | CIFAR10 | 81.55 / 49.36 |
|  | SVHN | 90.70 / 42.08 |
| VGG16 | CIFAR10 | 80.18 / 45.09 |
|  | SVHN | 89.41 / 45.71 |
| WRN28-4 | CIFAR10 | 83.68 / 50.12 |
|  | SVHN | 93.23 / 42.35 |
| ResNet50 | ImageNet | 60.25 / 36.82 |

### A.2  S2AP Finetuning

We defined the overall finetuning objective in Eq. 6 as:

$$\boldsymbol{w}^* \in \arg \min_{\boldsymbol{w}} \max_{\boldsymbol{\nu}} \mathcal{L}_r((\boldsymbol{w} + \boldsymbol{\nu}) \odot \boldsymbol{m}^*), \qquad (8)$$

$$\text{where } \|\boldsymbol{\nu}_l\| \leq \gamma \|\boldsymbol{w}_l\|, \qquad (9)$$

and $\gamma$ bounds the layer-wise perturbation and scales it based on each layer's weight magnitude, similarly to Wu et al. (2020). Hence, given the sparse parameterization defined by the mask $\boldsymbol{m}^*$ found during S2AP pruning in Algorithm 1, the S2AP finetuning formulation of Eq. 8 amounts to perturbing and updating only the non-zero (i.e., non-pruned) weights. While the S2AP procedure allows improving sharpness, stability, and robustness of the pruning mask per se, such a procedure enables aligning the finetuning objective with the pruning one and further improves robustness.

We provide a detailed implementation of the finetuning algorithm in Algorithm 2. Overall, the algorithm structure remains similar to Algorithm 1, with the only major variation that the perturbation $\boldsymbol{\nu}$ is applied on the non-zero weights $\boldsymbol{w} \odot \boldsymbol{m}^*$ only, instead of the entire score-space parameterized by $\boldsymbol{s}$.

**Algorithm 2:** Score-Sharpness-aware Adversarial Finetuning (S2AP Finetune).

**Input** : $\boldsymbol{w} \in \mathbb{R}^p$, pretrained weights; $\boldsymbol{m}^* \in \{0,1\}^p$, binary pruning mask; $\boldsymbol{x}$, training input samples; $\eta$, learning rate; $I$, number of iterations; $L$, number of layers; $\gamma$, perturbation scaling factor; $\hat{\mathcal{L}}$, robust loss.

**Output:** Finetuned weights $\boldsymbol{w}^* \in \mathbb{R}^p$

1 Initialize $\boldsymbol{\nu} \leftarrow 0$

2 **for** $i \leftarrow 1$ **to** $I$ **do**

3      Generate adversarial examples on pruned model $\boldsymbol{x}'_i \leftarrow \boldsymbol{x}_i + \boldsymbol{\delta}_i$

4      Compute robust loss $\hat{\mathcal{L}}(\boldsymbol{w} \odot \boldsymbol{m}^*) = \hat{\mathcal{L}}(\boldsymbol{w} \odot \boldsymbol{m}^*, \mathcal{D})$

5      Perturb pruned weights $\boldsymbol{\nu} \leftarrow \boldsymbol{\nu} + \eta \left( \nabla_{\boldsymbol{\nu}} \hat{\mathcal{L}}((\boldsymbol{w} + \boldsymbol{\nu}) \odot \boldsymbol{m}^*) / \|\nabla_{\boldsymbol{\nu}} \hat{\mathcal{L}}((\boldsymbol{w} + \boldsymbol{\nu}) \odot \boldsymbol{m}^*)\| \right)$

6      **for** $l \leftarrow 1$ **to** $L$ **do**

7          **if** $\|\boldsymbol{\nu}^{(l)}\| > \gamma \|\boldsymbol{w}^{(l)}\|$ **then**

8              Project $\boldsymbol{\nu}^{(l)} \leftarrow \left( \gamma \|\boldsymbol{w}^{(l)}\| / \|\boldsymbol{\nu}^{(l)}\| \right) \boldsymbol{\nu}^{(l)}$

9      Update weights: $\boldsymbol{w} \leftarrow \boldsymbol{w} - \eta \left( \nabla_{\boldsymbol{w}} \hat{\mathcal{L}}((\boldsymbol{w} + \boldsymbol{\nu}) \odot \boldsymbol{m}^*) / \|\nabla_{\boldsymbol{w}} \hat{\mathcal{L}}((\boldsymbol{w} + \boldsymbol{\nu}) \odot \boldsymbol{m}^*)\| \right)$

10      Restore weights: $\boldsymbol{w} \leftarrow \boldsymbol{w} - \boldsymbol{\nu}$

11 **return** $\boldsymbol{w}^* \leftarrow \boldsymbol{w}$

### A.3 $\gamma$-SELECTION

We select the $\gamma$ values, bounding the perturbation during S2AP pruning and finetuning, based on the adversarial robustness achieved choosing among a set of values $\gamma = \{0.00075, 0.001, 0.0025, 0.005, 0.0075, 0.01\}$. We show in Figure 5 the gamma search results for

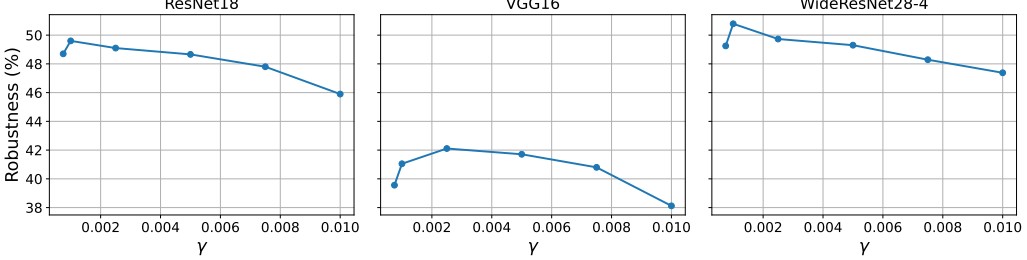

Figure 5: Robustness of S2AP pruning masks found using different $\gamma$ values bounding the score perturbation.

the CIFAR10 dataset, HARP method Zhao & Wressnegger (2023) at 90% sparsity. We repeat such an evaluation for each model/dataset combination at such sparsity, which we find descriptive of the trend on different sparsities as well, and find the best $\gamma$ value. Typically, we see a robustness increase for values prior to the best $\gamma$ found for the models (in this case 0.001 for ResNet18 and WideResNet28-4, and 0.0025 for VGG16), and then a corresponding robustness decrease after the best found $\gamma$.

### A.4 S2AP COMPUTATIONAL OVERHEAD

The S2AP procedure of Algorithm 1 inevitably induces a computational overhead. To provide an estimate of the required overhead, we report in Table 7 the time required by the original pruning methods (Orig.) and S2AP versions during pruning and average over the four sparsities. All experiments were conducted on a machine equipped with three NVIDIA RTX A6000 GPUs (48GB each), and the results of Table 7 were conducted on one of these 3 GPUs. Specifically, we report in Table 7 the results for CIFAR10 and SVHN models on 20 epochs (5 epochs for ImageNet) and batch size 128 without warm-up, thus allowing an equal comparison of original and S2AP procedures. Generally, we see an average increase in computing time of 15% circa, which, while it might be negligible

Table 7: S2AP overhead computation. We compute the time (hrs) required on a NVIDIA RTX A600 for each model/dataset combination, and report the average time required on different sparsities.

| Model | Dataset | Orig. (hrs) | S2AP (hrs) | Overhead (%) |
|---|---|---|---|---|
| ResNet18 | CIFAR10 | 3.27 | 3.96 | 17.42% |
| | SVHN | 4.91 | 5.21 | 5.75% |
| VGG16 | CIFAR10 | 1.41 | 1.73 | 18.49% |
| | SVHN | 2.43 | 2.75 | 11.63% |
| WRN28-4 | CIFAR10 | 6.12 | 6.97 | 12.20% |
| | SVHN | 6.77 | 7.31 | 7.38% |
| ResNet-50 | ImageNet | 15.08 | 17.11 | 13.46% |

in some application scenarios, still increases the overall computation. The same observation can be extended to ViT architectures.

## B ADDITIONAL EXPERIMENTS

We discuss here the additional experiments for S2AP. Precisely, we extend our approach to structured pruning, a standard "clean" pruning task, compare S2AP with AWP during the pruning stage, and finally analyze the effectiveness of S2AP on the common corruptions dataset.

Table 8: CIFAR-10 and SVHN results using RLTH with ResNet18, VGG-16, and WideResNet-28-4 across sparsity. Each cell shows clean/robust$_{\pm std}$ accuracy and the difference between Orig. and S2AP robust generalization gap ($\Delta$). In bold, the model with the highest robustness.

| Network | Sparsity | CIFAR-10 (RLTH) | | | SVHN (RLTH) | | |
|---|---|---|---|---|---|---|---|
| | | Orig. | S2AP | $\Delta$ | Orig. | S2AP | $\Delta$ |
| ResNet18 | 80% | 67.72 / 33.58 | **68.13 / 33.80** | +0.37 | 85.02 / 44.60 | **84.13 / 44.66** | +0.95 |
| | 90% | 69.32 / 34.42 | **69.30 / 34.92** | +0.52 | 83.65 / 44.07 | **84.51 / 44.50** | +1.29 |
| | 95% | 68.56 / 34.90 | **69.93 / 35.38** | +1.05 | 84.83 / 42.78 | **84.51 / 43.50** | +1.04 |
| | 99% | **66.19 / 32.66** | 60.27 / 31.09 | -4.45 | 81.72 / 41.59 | **80.61 / 41.73** | +1.25 |
| VGG16 | 80% | 18.63 / 11.04 | **22.62 / 12.20** | +1.19 | 32.89 / 18.70 | **32.83 / 19.01** | +0.37 |
| | 90% | 23.36 / 13.17 | **24.63 / 13.19** | +1.01 | **34.21 / 18.78** | 37.20 / 17.31 | -1.92 |
| | 95% | 30.04 / 12.06 | **26.62 / 19.33** | +6.69 | **37.29 / 20.06** | 34.40 / **21.86** | +2.71 |
| | 99% | **18.36 / 14.47** | 19.09 / 12.50 | -1.70 | **20.06 / 19.68** | 21.68 / 18.00 | -2.30 |
| WRN28-4 | 80% | 68.94 / 34.55 | **69.71 / 34.91** | +0.65 | 87.82 / 44.52 | **87.81 / 44.71** | +0.36 |
| | 90% | 70.05 / 33.53 | **69.65 / 34.39** | +0.86 | 88.83 / 43.95 | **85.97 / 44.53** | +1.58 |
| | 95% | **69.29 / 34.40** | 68.69 / 33.55 | -0.75 | **86.82 / 43.85** | 88.13 / 42.95 | -1.56 |
| | 99% | 63.19 / 29.56 | **62.13 / 29.83** | +0.89 | 77.29 / 36.68 | **80.80 / 37.96** | +3.79 |

### B.1 EXPERIMENTS ON RLTH

As in Table 1 and Table 2, for the HARP and HYDRA methods, RLTH can benefit from robustness increases from the S2AP method, as we show in Table 8. This result is not obvious, as RLTH involves a different pruning pipeline than existing methods. As opposed to starting from a pretrained model, pruning, and then finetuning, such method in fact follows the lottery ticket hypothesis Frankle & Carbin (2019), which admits the existence of subnetworks within dense, randomly initialized models. Overall, compared to other methods, we see RLTH pruned models having lower accuracies due to the pruned random initialization and absence of finetuning. The improved robustness of S2AP, considering the absence of finetuning on RLTH, further corroborates to the ablation study discussed in Table 5, which shows how S2AP, independently from finetuning at all, is capable of reaching higher adversarial robustness from pruning already.

Table 9: Channel Pruning with S2AP on CIFAR10 dataset.

| Network | Sparsity (%) | HARP-Orig. | S2AP-HARP | HYDRA-Orig. | S2AP-HYDRA |
|---|---|---|---|---|---|
| ResNet18 | 4 | 49.60 | **50.36** | 49.32 | **50.85** |
|  | 15 | 48.28 | **48.63** | 38.69 | **39.79** |
| VGG-16 | 4 | 47.37 | **48.18** | 47.02 | 47.33 |
|  | 15 | **37.38** | 37.17 | 33.15 | **34.53** |

## B.2 Experiments on Structured Pruning

Unstructured pruning serves as a great mathematical prototype for neural networks, allowing for single weights to be pruned. Empirically, this is widely accepted as an upper-bound on the other important category of pruning methods, i.e., *structured pruning* Liu & Wang (2023). From a practical perspective, structured pruning allows for removing entire network structures, such as channels and filters, and constitutes a readily usable network size reduction. In fact, while unstructured pruning requires a still maturing dedicated hardware, structured pruning implies reducing network size and leveraging it directly Liu & Wang (2023). To validate the effectiveness of our S2AP method, given the high relevance of structured pruning methods, we extend, in Table 9, experiments of both HARP and HYDRA methods to channel pruning, relying on the ResNet18 and VGG16 networks on CIFAR10 as a testbed. Instead of the classic sparsity rate $k$, for channel pruning we refer to the reduction in floating point operations (FLOPs). Specifically, we obtain 4 or 15 times fewer FLOPs than the original dense model, thus improving the overall model efficiency and computing time. Such a form of sparsity is more compatible with standard hardware acceleration and better suited for real-world deployment. Overall, these results confirm that S2AP generalizes effectively also to different kinds of pruning structures, further reinforcing the versatility of our approach.

Table 10: Mask clean / robust accuracy (mean$_{\pm\text{std}}$) on CIFAR10 and SVHN across sparsity levels using ResNet18, VGG-16, and WideResNet-28-4.

| Network | Sparsity | CIFAR10 | | | | SVHN | | | |
|---|---|---|---|---|---|---|---|---|---|
| | | HARP | | HYDRA | | HARP | | HYDRA | |
| | | Orig. | S2AP | Orig. | S2AP | Orig. | S2AP | Orig. | S2AP |
| ResNet18 | 80% | 83.19/48.88$_{\pm0.73}$ | **82.03/49.55**$_{\pm0.69}$ | 82.13/48.56$_{\pm0.66}$ | **82.87/48.98**$_{\pm0.75}$ | 90.10/46.56$_{\pm0.66}$ | **87.74/49.18**$_{\pm0.77}$ | 90.63/45.74$_{\pm0.73}$ | **89.80/46.11**$_{\pm0.69}$ |
| | 90% | 82.98/49.42$_{\pm0.72}$ | **83.12/49.60**$_{\pm0.74}$ | 80.55/47.41$_{\pm0.84}$ | **82.26/48.06**$_{\pm0.71}$ | 90.20/49.04$_{\pm0.79}$ | 90.17/48.28$_{\pm0.81}$ | 84.82/45.61$_{\pm0.70}$ | **88.77/47.62**$_{\pm0.83}$ |
| | 95% | **82.26/49.04**$_{\pm0.76}$ | 82.48/48.43$_{\pm0.78}$ | 78.98/45.55$_{\pm0.91}$ | **79.47/45.61**$_{\pm0.86}$ | 92.22/41.66$_{\pm1.21}$ | **89.07/45.58**$_{\pm0.75}$ | 83.57/44.53$_{\pm0.84}$ | **88.82/45.14**$_{\pm0.72}$ |
| | 99% | 72.97/40.99$_{\pm1.34}$ | **75.46/41.86**$_{\pm1.19}$ | 69.63/35.15$_{\pm1.48}$ | **69.66/36.74**$_{\pm1.42}$ | 85.01/40.79$_{\pm0.94}$ | **85.35/45.77**$_{\pm1.07}$ | 83.12/40.85$_{\pm0.99}$ | 79.19/37.93$_{\pm1.22}$ |
| VGG-16 | 80% | 75.78/41.93$_{\pm0.82}$ | **76.51/42.84**$_{\pm0.85}$ | 74.86/40.31$_{\pm0.95}$ | **76.15/41.39**$_{\pm0.91}$ | 89.51/46.95$_{\pm0.78}$ | **89.53/48.93**$_{\pm0.84}$ | 85.67/45.78$_{\pm0.89}$ | **87.39/46.17**$_{\pm0.75}$ |
| | 90% | 73.89/41.69$_{\pm0.86}$ | **75.86/42.11**$_{\pm0.87}$ | 73.19/38.12$_{\pm1.12}$ | **75.17/40.61**$_{\pm0.93}$ | 89.73/47.30$_{\pm0.79}$ | 87.12/46.28$_{\pm0.76}$ | 84.91/44.22$_{\pm0.81}$ | **87.39/46.17**$_{\pm0.88}$ |
| | 95% | **73.55/40.21**$_{\pm0.97}$ | 74.68/39.13$_{\pm0.99}$ | 62.30/31.81$_{\pm1.42}$ | **72.86/38.03**$_{\pm1.08}$ | 87.86/46.51$_{\pm0.75}$ | **87.85/47.96**$_{\pm0.71}$ | 82.07/42.43$_{\pm0.84}$ | **85.47/43.78**$_{\pm0.73}$ |
| | 99% | 52.59/24.22$_{\pm1.52}$ | **72.76/36.41**$_{\pm1.21}$ | 40.77/20.54$_{\pm1.68}$ | **60.91/29.67**$_{\pm1.49}$ | 84.70/43.42$_{\pm0.77}$ | 84.07/43.91$_{\pm0.81}$ | 79.75/31.06$_{\pm1.34}$ | **83.82/32.64**$_{\pm1.41}$ |
| WRN28-4 | 80% | 82.97/50.45$_{\pm0.81}$ | **83.24/50.59**$_{\pm0.73}$ | 82.59/50.31$_{\pm0.78}$ | **83.05/50.41**$_{\pm0.76}$ | 90.91/43.79$_{\pm0.74}$ | **88.90/47.02**$_{\pm0.78}$ | 90.49/49.43$_{\pm0.73}$ | 88.87/47.50$_{\pm0.71}$ |
| | 90% | 81.82/50.56$_{\pm0.77}$ | **82.66/50.79**$_{\pm0.72}$ | 80.71/47.75$_{\pm0.88}$ | **81.92/49.30**$_{\pm0.80}$ | 91.41/45.89$_{\pm0.75}$ | **90.61/46.31**$_{\pm0.74}$ | 90.83/43.80$_{\pm0.76}$ | **89.51/45.66**$_{\pm0.78}$ |
| | 95% | 80.44/49.07$_{\pm0.91}$ | **80.82/49.37**$_{\pm0.87}$ | 79.82/46.97$_{\pm0.97}$ | 80.19/46.85$_{\pm0.93}$ | 88.43/41.69$_{\pm0.79}$ | **85.74/45.41**$_{\pm0.76}$ | 88.35/48.01$_{\pm0.75}$ | **88.61/48.35**$_{\pm0.73}$ |
| | 99% | 71.57/38.89$_{\pm1.39}$ | **71.64/39.89**$_{\pm1.22}$ | 70.33/34.57$_{\pm1.47}$ | **71.40/36.30**$_{\pm1.34}$ | 84.99/43.58$_{\pm0.78}$ | 80.51/40.87$_{\pm0.81}$ | 85.54/40.57$_{\pm0.79}$ | 84.32/38.84$_{\pm0.82}$ |

## B.3 Experiments on Standard Clean Pruning

On several occasions throughout the paper, we remarked on the generality of the S2AP method beyond the specific adversarial pruning task. We thus aim to first confirm the S2AP effectiveness and utility on the most basic task required by such networks: standard classification. Hence, we prune networks using a standard cross-entropy loss, disregarding the adversarial robustness objective, and fine-tune accordingly. We show the results of such experiments in Table 11, where we reveal how S2AP improves not only adversarial robustness, but also clean accuracy on a standard classification task for the CIFAR10 dataset. We thus confirm the initial claim of general use and applicability of S2AP to different tasks and scenarios, not limited to the adversarial pruning case.

Furthermore, we extend the results reported in Table 5 with the corresponding clean accuracy values. In Table 10, we confirm the same trends observed for robustness. Finally, we specify that the $\Delta$ quantity is used in our analysis as a marker of whether improving robustness comes at the cost of noticeably degrading clean accuracy. In the adversarial robustness literature, it is common for robustness-oriented methods to introduce a trade-off between clean and robust accuracy, meaning that gains in adversarial robustness are obtained at the expense of significantly lower clean accuracy.

Table 11: Clean accuracy (%) under different sparsity levels. For each pruning method (HARP/HYDRA), we report Orig. and S2AP variants. Bold indicates the best between Orig. and S2AP.

| Network | Sparsity (%) | HARP-Orig. | S2AP-HARP | HYDRA-Orig. | S2AP-HYDRA |
|---------|--------------|------------|-----------|-------------|------------|
| ResNet18 | 80 | 94.70 | **94.85** | **94.90** | 94.61 |
| | 90 | 94.12 | **94.89** | 94.37 | **94.73** |
| | 95 | 93.18 | **94.56** | 94.20 | **94.84** |
| | 99 | 92.27 | **93.01** | 90.22 | **90.38** |
| VGG-16 | 80 | 92.17 | **92.82** | 92.46 | **93.20** |
| | 90 | 92.34 | **92.99** | 92.52 | **93.70** |
| | 95 | 92.41 | **93.03** | 91.41 | **91.95** |
| | 99 | **90.96** | 91.76 | 87.32 | **87.40** |

Table 12: Robust accuracy (%) on CIFAR-10-C under different sparsity levels. Bold indicates the best between Orig. and S2AP for each pruning method.

| Network | Sparsity (%) | HARP-Orig. | S2AP-HARP | HYDRA-Orig. | S2AP-HYDRA |
|---------|--------------|------------|-----------|-------------|------------|
| ResNet18 | 80 | 72.52 | **73.08** | 71.75 | **72.01** |
| | 90 | 72.62 | **73.12** | 71.54 | **72.16** |
| | 95 | 72.27 | **73.23** | 70.02 | **70.59** |
| | 99 | **68.52** | 68.48 | 65.41 | **66.50** |
| VGG-16 | 80 | 70.07 | **70.97** | 68.84 | **68.98** |
| | 90 | 71.15 | **71.34** | 69.23 | 68.71 |
| | 95 | 69.97 | **70.05** | 68.15 | **68.33** |
| | 99 | **66.89** | 67.45 | 59.09 | **59.26** |
| WRN | 80 | 72.73 | **72.88** | 72.59 | **73.54** |
| | 90 | 72.54 | **73.06** | 71.75 | **73.08** |
| | 95 | 73.03 | **73.32** | 72.83 | **72.85** |
| | 99 | 67.63 | **67.95** | 65.61 | **66.04** |

We track that with $\Delta = (\text{acc}_{\text{Orig.}} - \text{robustness}_{\text{Orig.}}) - (\text{acc}_{\text{S2AP}} - \text{robustness}_{\text{S2AP}})$. Hence, a positive $\Delta$ implies that S2AP's gap is smaller than Orig.'s gap. We consistently find this quantity to be positive.

## B.4 EXPERIMENTS ON CORRUPTIONS

Following on from the previous experiments, extending to standard pruning, it is likewise relevant to consider further tasks. We thus choose to test on the general robustness to corruption task by including experiments on the CIFAR10-C dataset. We select a corruption severity of 3, and show the results in Table 12. As in previous experiments, we demonstrate how S2AP is further applicable to different tasks and keeps its superiority compared to other methods. We thus believe that such an extension corroborates the claims and results obtained in adversarial robustness, besides broadening the method's applicability.

## B.5 PERTURBING WEIGHTS OR SCORES?

One of the big novelties that can be found in S2AP is the focus on the score-space, rather than the usual weight-space where prior sharpness-minimization approaches focused in the past. In turn, a natural question is whether sharpness minimization should be performed in weight space, as done in prior work such as Adversarial Weight Perturbations (AWP), or in score space, as we propose in S2AP. In adversarial pruning, the pruning mask is determined by the ranking of importance scores rather than the weights themselves. Hence, perturbing scores directly addresses the variables that drive mask selection, potentially stabilizing the top-k cutoff. While this intuition suggests a better alignment with the pruning objective, our main justification is empirical. As shown in Table 13,

Table 13: ResNet18 on CIFAR-10: accuracy (%) under different sparsity levels when pruning with AWP (perturbing weights) vs. S2AP (perturbing scores). Bold indicates the best between AWP and S2AP for each method.

| Network | Sparsity (%) | HARP-AWP | HARP-S2AP | HYDRA-AWP | HYDRA-S2AP |
|---------|-------------|----------|-----------|-----------|------------|
| ResNet18 | 80 | 47.32 | **49.55** | 46.12 | **48.98** |
| | 90 | 47.80 | **49.60** | 45.19 | **48.06** |
| | 95 | 46.91 | **48.43** | 42.77 | **45.61** |
| | 99 | 40.35 | **41.86** | 34.34 | **36.74** |

perturbing scores during mask search consistently leads to higher robust accuracy than perturbing weights, across different networks and datasets. These results, which indicate the mask robustness before finetuning as in Table 5, indicate that score-space perturbations are more effective at preserving robustness in adversarial pruning than their weight-space counterparts. While a more formal reason describing the differences between applying AWP or S2AP during pruning is missing, we believe that a role behind the greater success of score perturbations could also be played by the increased mask stability.

## C   MEASURING SCORE-SPACE SHARPNESS AND MASK STABILITY

We measure score-space sharpness relying on two specific approaches: the largest eigenvalue computation $\lambda_{max}$ and the loss difference (following Stutz et al. (2021); Andriushchenko et al. (2023)). We dedicate this section to describing both approaches in detail, and provide additional experiments and results on more model and dataset combinations. In addition to minimizing sharpness, however, S2AP also improves the mask stability during pruning. In turn, we conclude this section by describing the proposed measure in detail and showing additional experiments.

### C.1   MEASURING LARGEST EIGENVALUE

To compute the largest eigenvalue of the Hessian $\nabla_s^2 \mathcal{L}_r(s)$ with respect to the score parameters, we adopt the classical power iteration method. Starting from a random unit-norm vector $v^{(0)} \in \mathbb{R}^p$, we iteratively compute:

$$v^{(t+1)} = \frac{\nabla_s^2 \mathcal{L}_r(s) v^{(t)}}{\|\nabla_s^2 \mathcal{L}_r(s) v^{(t)}\|_2}, \tag{10}$$

where $\mathcal{L}_r(s) = \mathcal{L}_r(w \odot M(s, k), \mathcal{D})$ is the robust loss, that we denote as $\mathcal{L}_r(s)$ to lighten notation. After $T$ iterations, we compute the Rayleigh quotient as an approximation of the largest eigenvalue:

$$\lambda_{\max} \approx \left\langle v^{(T)}, \nabla_s^2 \mathcal{L}_r(s) v^{(T)} \right\rangle. \tag{11}$$

We select $T = 10$ iterations to compute the quotient, and specify that we implement this computation using Hessian-vector products via automatic differentiation, thus refraining from explicitly forming the Hessian Jastrzębski et al. (2017). This procedure is run at each pruning iteration of both the S2AP and original methods. We then average the resulting $\lambda_{max}$ values across each iteration and plot the corresponding sharpness trends against epochs. While we show the CIFAR10 HARP method for ResNet18 in Figure 1c and for WideResNet28-4 in Figure 2, we complete the remaining plots from Figure 8 to Figure 17. Overall, the plots show how methods pruned with S2AP hold, apart from a few exceptions, a consistently lower maximum eigenvalue across multiple architectures, datasets, pruning methods, and sparsities. We specify how, on the first few epochs, the resulting $\lambda_{max}$ has a negligible difference between Orig. and S2AP methods (hence the first 10 warped epochs).

### C.2   MEASURING SCORE-SPACE LOSS DIFFERENCE

Measuring sharpness through a loss difference requires perturbing a "reference" loss value $\mathcal{L}_r(w \odot M(s, k))$, representing a local minima, through a perturbation $\nu$ which enables measuring sharpness as follows:

$$\max_{\|\nu \odot c^{-1}\|_\infty \leq \rho} \mathcal{L}_r(w \odot M(s + \nu, k), \mathcal{D}) - \mathcal{L}_r(w \odot M(s, k), \mathcal{D}) \tag{12}$$

where $c$ is a positive scaling vector used to make the sharpness definition reparameterization-invariant, addressing the well-known problems of sharpness measures Dinh et al. (2017), and the operator $\odot /^{-1}$ defines element-wise multiplication/inversion. We specify that such a formulation corresponds to the one presented in Andriushchenko et al. (2023), yet adapted to our score-space case. Overall, we thus perturb the score-space and measure the corresponding loss variation imposed by the shift and mask variation, which we expect to be lower in the S2AP case.

In our experiments, we evaluate different $\rho$ values, and show in Table 14 an overview of the CIFAR10 results. Overall, we see how S2AP consistently reduces sharpness, except for some specific cases at high sparsities. In this regard, however, increasing the corresponding $\rho$ value appears to still favor S2AP, suggesting that lower values might not be enough (hence, we choose $\rho = 0.01$ in the plot of Figure 3).

### C.3 MASK STABILITY

We measure mask stability based on the Hamming distance $h$, which equals measuring the rate of change between masks as follows:

$$h = \|\boldsymbol{m}_0 \oplus \boldsymbol{m}_t\|_1 / |\boldsymbol{m}_0|, \text{ where } t \in \{1, 2, \ldots, T\}, \tag{13}$$

where $\boldsymbol{m}_t$ represents the mask found at epoch $t$, $\oplus$ is the XOR operator measuring the number of differing bits, and $T$ is the total number of epochs. We compute $\boldsymbol{h} = \{h_1, h_2, \ldots, h_T\}$, thus measuring the distance from the first mask in each epoch, for both original (Orig.) and S2AP adversarial pruning methods. Overall, lower $h$ values indicate improved stability, as the number of changed selected weights is, in turn, lower. To provide a useful analysis, we compute two vectors, $\boldsymbol{h}_{orig}$ and $\boldsymbol{h}_{S2AP}$, by saving the masks at each epoch while pruning, that we then subtract as $\boldsymbol{h}_{orig} - \boldsymbol{h}_{S2AP}$. Hence, we obtain a single curve plot that, when positive, indicates that the S2AP method is more stable than the original one, and vice versa when negative.

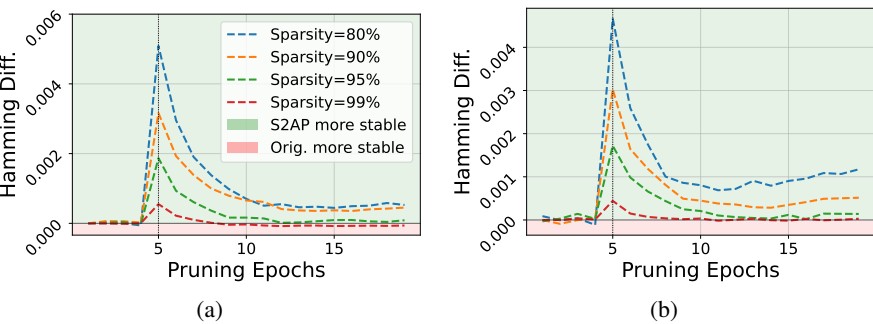

(a)                                 (b)

Figure 6: Improved mask stability of Resnet18 (a) and WideResNet28-4 (b) on the HYDRA method.

Stability is depicted for CIFAR10 HARP method and ResNet18 in Figure 1b, and for WideResNet28-4 in Figure 4. Nonetheless, we provide additional plots for the remaining combinations in Figure 6, where we show the improved mask stability of S2AP on the HYDRA method as well. For VGG16 models, interestingly, we find the stability trend often favors the Orig. models instead of S2AP, particularly at lower sparsity. We analyze such a result through the plots of Figure 7. Overall, such a measure allows assessing how much the pruning decisions evolve over time relative to their starting point.

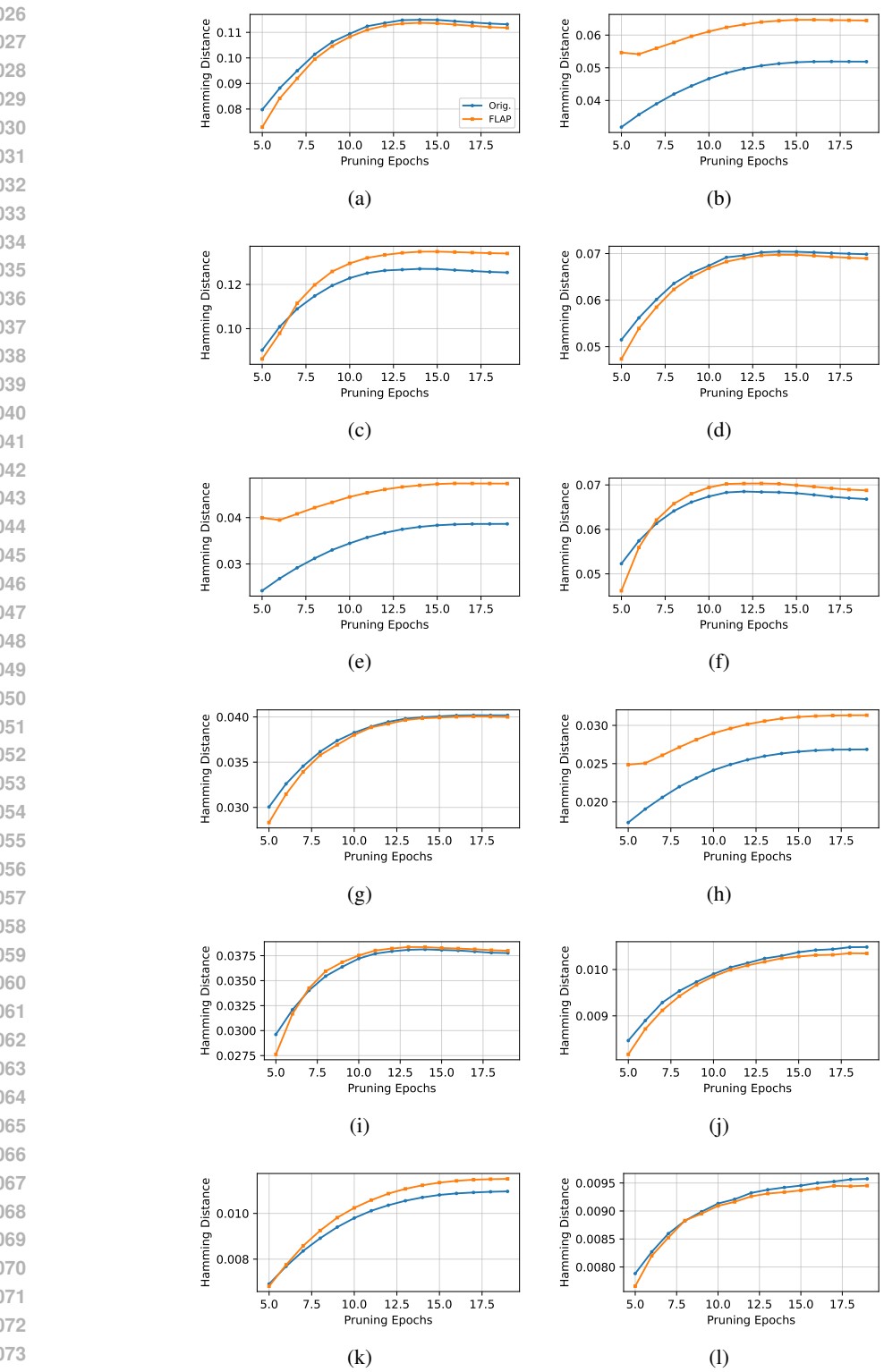

Figure 7: Single Hamming distances of VGG16 on CIFAR10 and SVHN after the first 5 pruning epochs. In (a), (b), and (c) the 80% sparsity for HARP on CIFAR10, HYDRA on CIFAR10, and HARP on SVHN; in (d), (e), and (f) the 90% sparsity for HARP on CIFAR10, HYDRA on CIFAR10, and HARP on SVHN; in (g), (h), and (i) the 95% sparsity for HARP on CIFAR10, HYDRA on CIFAR10, and HARP on SVHN; and in (j), (k), and (l) the 99% sparsity for HARP on CIFAR10, HYDRA on CIFAR10, and HARP on SVHN.

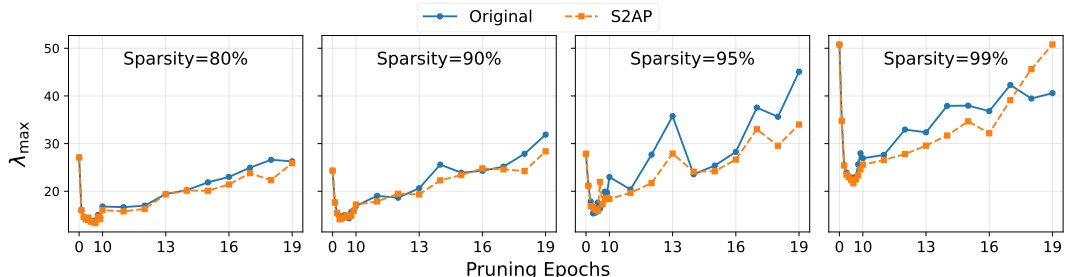

Figure 8: Largest eigenvalue across HYDRA pruning epochs for ResNet18 on CIFAR10.

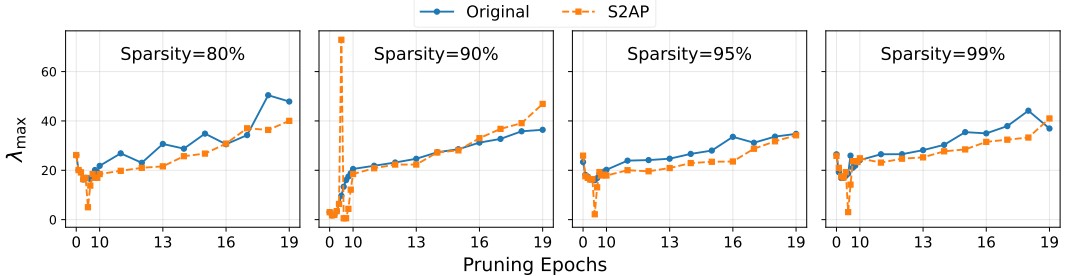

Figure 9: Largest eigenvalue across HARP pruning epochs for ResNet18 on SVHN.

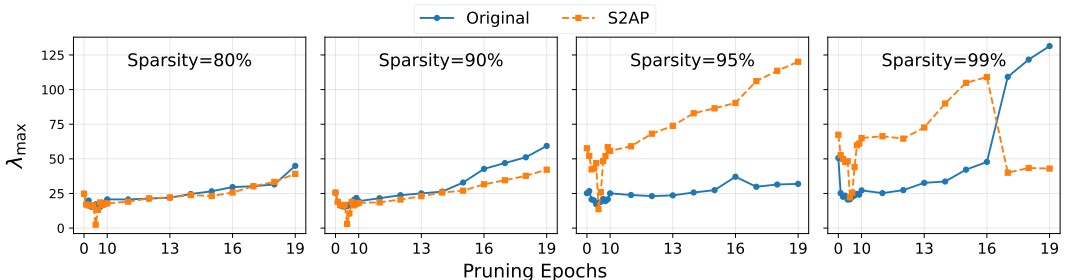

Figure 10: Largest eigenvalue across HYDRA pruning epochs for ResNet18 on SVHN.

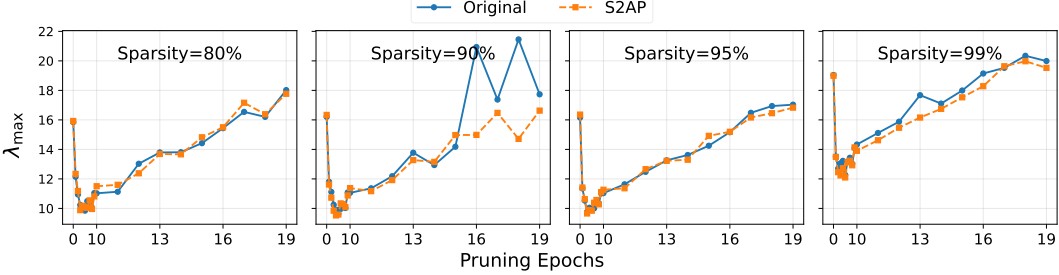

Figure 11: Largest eigenvalue across HARP pruning epochs for VGG on CIFAR10.

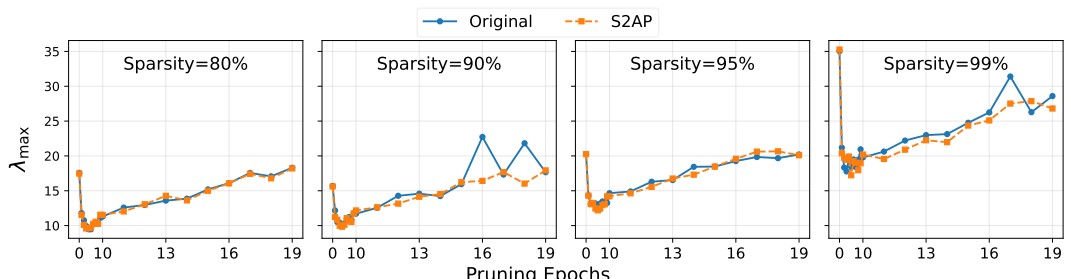

Figure 12: Largest eigenvalue across HYDRA pruning epochs for VGG on CIFAR10.

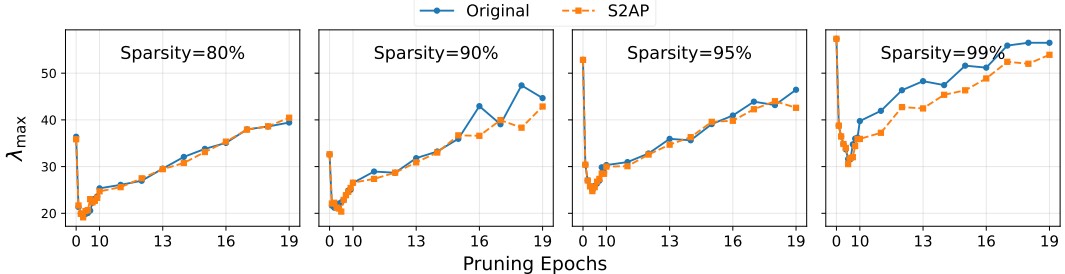

Figure 13: Largest eigenvalue across HARP pruning epochs for VGG on SVHN.

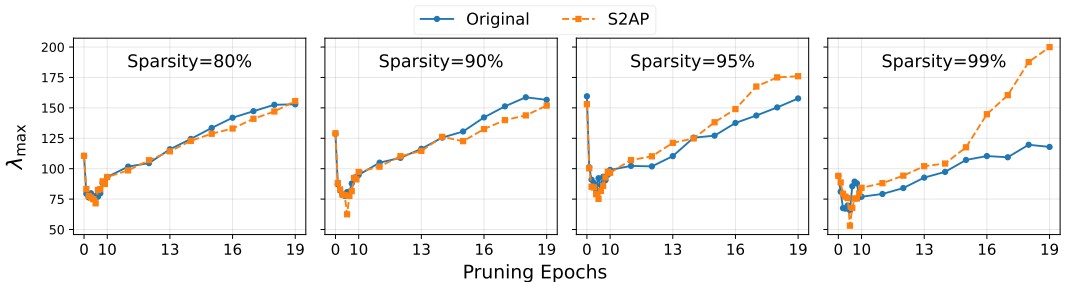

Figure 14: Largest eigenvalue across HYDRA pruning epochs for VGG16 on SVHN.

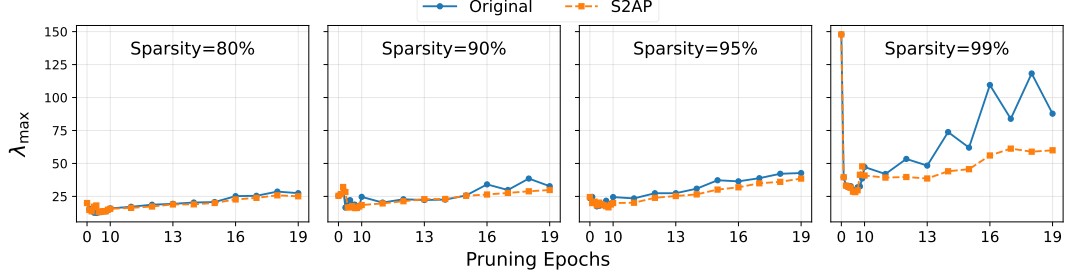

Figure 15: Largest eigenvalue across HYDRA pruning epochs for WideResNet28-4 on CIFAR10.

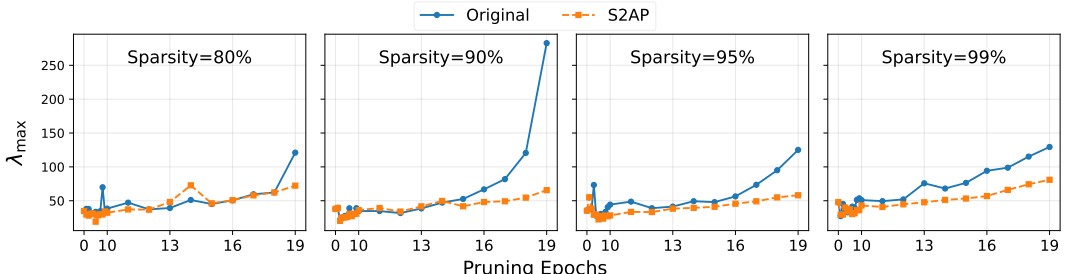

Figure 16: Largest eigenvalue across HARP pruning epochs for WideResNet28-4 on SVHN.

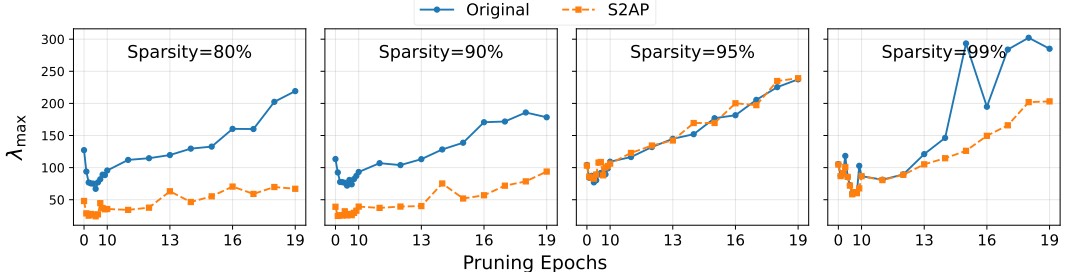

Figure 17: Largest eigenvalue across HYDRA pruning epochs for WideResNet28-4 on SVHN.

Table 14: CIFAR10 Sharpness comparison across sparsity levels and $\rho$ values using Orig. and S2AP pruning strategies. Lower sharpness values are in **bold**.

| Model | Sparsity (%) | $\rho$ | HARP | | HYDRA | |
|---|---|---|---|---|---|---|
| | | | Orig. | S2AP | Orig. | S2AP |
| ResNet18 | 80 | 0.001 | 0.08316 | **0.07723** | **0.08820** | 0.09274 |
| | | 0.0025 | 0.10498 | **0.09742** | **0.11074** | 0.11315 |
| | | 0.005 | 0.14170 | **0.13142** | 0.14845 | **0.14702** |
| | | 0.0075 | 0.18016 | **0.16670** | 0.18784 | **0.18171** |
| | | 0.01 | 0.22096 | **0.20322** | 0.22839 | **0.21794** |
| | 90 | 0.001 | 0.07001 | **0.06848** | 0.08675 | **0.07637** |
| | | 0.0025 | 0.08566 | **0.08329** | 0.10258 | **0.09097** |
| | | 0.005 | 0.11239 | **0.10844** | 0.13116 | **0.11596** |
| | | 0.0075 | 0.14069 | **0.13311** | 0.15879 | **0.14189** |
| | | 0.01 | 0.16937 | **0.15981** | 0.18928 | **0.16741** |
| | 95 | 0.001 | 0.06409 | **0.06346** | 0.07383 | **0.06957** |
| | | 0.0025 | 0.07676 | **0.07504** | 0.08597 | **0.08035** |
| | | 0.005 | 0.09787 | **0.09401** | 0.10601 | **0.09822** |
| | | 0.0075 | 0.11952 | **0.11332** | 0.12648 | **0.11706** |
| | | 0.01 | 0.14170 | **0.13420** | 0.14774 | **0.13521** |
| | 99 | 0.001 | **0.05921** | 0.06428 | 0.05573 | **0.05148** |
| | | 0.0025 | **0.06637** | 0.07114 | 0.06233 | **0.05728** |
| | | 0.005 | **0.07810** | 0.08258 | 0.07365 | **0.06711** |
| | | 0.0075 | **0.08930** | 0.09392 | 0.08483 | **0.07718** |
| | | 0.01 | 0.10852 | **0.10082** | 0.09631 | **0.08761** |
| VGG16 | 80 | 0.001 | 0.05925 | **0.05837** | **0.05547** | 0.05604 |
| | | 0.0025 | 0.07916 | **0.07852** | **0.07089** | 0.07134 |
| | | 0.005 | 0.11328 | **0.11260** | **0.09726** | 0.09804 |
| | | 0.0075 | 0.14856 | **0.14736** | **0.12450** | 0.12492 |
| | | 0.01 | 0.18579 | **0.18412** | 0.15311 | **0.15298** |
| | 90 | 0.001 | **0.05429** | 0.05503 | 0.05616 | **0.05531** |
| | | 0.0025 | **0.07059** | 0.07160 | 0.06881 | **0.06749** |
| | | 0.005 | **0.09846** | 0.09970 | 0.08966 | **0.08804** |
| | | 0.0075 | **0.12649** | 0.12834 | 0.11178 | **0.10993** |
| | | 0.01 | **0.15518** | 0.15657 | 0.13426 | **0.13282** |
| | 95 | 0.001 | 0.05003 | **0.04989** | 0.05386 | **0.04861** |
| | | 0.0025 | 0.06286 | **0.06237** | 0.06358 | **0.05832** |
| | | 0.005 | 0.08452 | **0.08361** | 0.08018 | **0.07514** |
| | | 0.0075 | 0.10634 | **0.10562** | 0.09726 | **0.09266** |
| | | 0.01 | 0.12874 | **0.12760** | 0.11470 | **0.11020** |
| | 99 | 0.001 | 0.04367 | **0.04174** | 0.04815 | **0.04601** |
| | | 0.0025 | 0.05087 | **0.04909** | 0.05509 | **0.05261** |
| | | 0.005 | 0.06309 | **0.06165** | 0.06656 | **0.06362** |
| | | 0.0075 | 0.07565 | **0.07486** | 0.07853 | **0.07477** |
| | | 0.01 | **0.08851** | 0.08857 | 0.09058 | **0.08608** |
| WRN | 80 | 0.001 | 0.07913 | **0.07880** | 0.07991 | **0.07489** |
| | | 0.0025 | 0.09723 | **0.09593** | 0.09703 | **0.09096** |
| | | 0.005 | 0.12753 | **0.12433** | 0.12677 | **0.11843** |
| | | 0.0075 | 0.15926 | **0.15295** | 0.15632 | **0.14677** |
| | | 0.01 | 0.19145 | **0.18250** | 0.18822 | **0.17620** |
| | 90 | 0.001 | 0.07006 | **0.06571** | 0.07159 | **0.07602** |
| | | 0.0025 | 0.08337 | **0.07786** | 0.08506 | **0.08826** |
| | | 0.005 | 0.10571 | **0.09843** | 0.10735 | **0.10935** |
| | | 0.0075 | 0.12777 | **0.11919** | 0.13010 | **0.13164** |
| | | 0.01 | 0.15019 | **0.13998** | 0.15363 | **0.15292** |
| | 95 | 0.001 | **0.05809** | 0.06162 | 0.06952 | **0.06229** |
| | | 0.0025 | **0.06847** | 0.07097 | 0.07998 | **0.07141** |
| | | 0.005 | **0.08537** | 0.08655 | 0.09721 | **0.08766** |
| | | 0.0075 | **0.10154** | 0.10193 | 0.11500 | **0.10338** |
| | | 0.01 | 0.11772 | **0.11709** | 0.13249 | **0.11976** |
| | 99 | 0.001 | **0.05823** | 0.05911 | 0.04382 | 0.04925 |
| | | 0.0025 | **0.06376** | 0.06482 | 0.05483 | **0.04949** |
| | | 0.005 | **0.07283** | 0.07452 | 0.06425 | **0.05900** |
| | | 0.0075 | **0.08122** | 0.08487 | 0.07172 | **0.06888** |
| | | 0.01 | **0.08968** | 0.09446 | 0.08512 | **0.07535** |