# OpenReview forum: "S2AP: Score-space Sharpness Minimization for Adversarial Pruning"
_ICLR.cc/2026/Conference — Submitted to ICLR 2026_

### Official Review · Reviewer_ASif · 2025-10-25

**Soundness:** 3
**Presentation:** 3
**Contribution:** 2
**Rating:** 4
**Confidence:** 4

**Summary:**

This paper introduces S2AP, a plug-in method for adversarial pruning that minimizes sharpness in the score-space—the space of importance scores used to select which weights to prune. By perturbing these scores during mask optimization, S2AP stabilizes the pruning process, reduces sensitivity to small score variations, and consistently improves adversarial robustness across various models, datasets, and sparsity levels, without altering the core logic of existing pruning methods.

**Strengths:**

+ The visualizations of the experimental results are rich and nicely done.

+ The workflow of the proposed S2AP method is clearly presented and easy to understand.

**Weaknesses:**

+ The motivation needs to be further clarified. In particular, why the score-space optimization can lead to sharp local minima in the loss landscape? Additional empirical results or theoretical analysis are required to support this claim.

+ The flatness of the robust loss landscape has been already explored to improve adversarial robustness (e.g., Ref. [1]). What is the difference between S2AP and existing studies in terms of measuring the flatness of the robust loss landscape?

+ Another major concern is about the computational cost. It seems that S2AP introduces several time-consuming update processes during the pruning. A comprehensive empirical study regarding the computational cost would help the reader better understand the cost of the S2AP method.

[1] AdvRush: Searching for adversarially robust neural architectures, ICCV 2021.

**Questions:**

Please refer to the Weaknesses part.

---

> ### Author Response · Authors · 2025-11-21
> **Response by authors**
>
> We thank the reviewer for their feedback, and for recognizing the value of the efforts behind results presentation and paper clarity. We respond to each of the listed weaknesses below.
>
> **W1 - Why score-space optimization leads to sharp local minima?** We thank the reviewer for pointing out the need for clarification.
> We believe that while standard weight optimization can suffer from sharp minima, the issue is exacerbated in score-space pruning due to the continuous-to-discrete mapping. In fact, importance scores $\mathbf s$ are continuous variables optimized to determine a binary mask via a hard `top-k` threshold. In a sharp minimum, the loss landscape is highly sensitive to small variations in $\mathbf s$, because a tiny perturbation in score-space (typical of a sharp minimum) potentially triggers a massive structural change in the pruning binary mask (flipping 0s and 1s). This makes the resulting sparse architecture search brittle and volatile in the presence of a sharp minimum. S2AP addresses this by explicitly finding a flat region where score variations do not significantly alter the mask.
>
> Regarding evidence, we strongly agree. In fact, we rigorously measure score-space sharpness using two distinct metrics in Section 4.3: (i) the largest Hessian eigenvalues with respect to the scores using power iteration ($\lambda_{max}$) and (ii) a perturbation-based loss difference, following prior work. We then show the direct consequence of this sharpness via the Hamming distance in Figures 1b and 4. We demonstrate that sharp minima of the original methods lead to higher mask volatility (measured as the Hamming distance between masks), whereas S2AP results in a more stable mask selection.
>
> **W2 - Difference from referenced paper:**  In the referenced paper, the AdvRush authors measure sharpness with respect to the input loss landscape (perturbing the $x$ in $\mathcal{L}(f(\mathbf x; \mathbf w), y)$). In contrast, in the context of S2AP, we measure sharpness in the score-space landscape, where the variables are the importance scores $\mathbf{s}$ used to generate the pruning mask via $M(\mathbf{s},k)$. This is conceptually closer to the typical weight-space sharpness measure (perturbations of $\mathbf w$), being the scores optimized continuously. However, score-space sharpness is fundamentally different from weight-space sharpness due to the continuous-to-discrete `top-k` cutoff, with small changes in $\mathbf{s}$ causing discrete mask changes. This directly affects mask stability and is precisely what S2AP targets.
>
> **W3 - Cost:** We thank the reviewer for raising this point. We agree that understanding the overhead is crucial for any method aimed at minimizing sharpness. We are happy to clarify that we have already conducted a comprehensive empirical study regarding computational cost, which is detailed in Appendix A.4 and Table 7.
> In particular, we show that the additional cost is modest (approximately 15%) and remains stable across various architectures and sparsity levels.
>
> **Final remarks:**
> We appreciate the reviewer’s comments. Our clarifications and empirical evidence address the comments on motivation, related work, and cost, and we hope they help confirm the noted significance of S2AP’s contribution.

---

> > ### Comment · Reviewer_ASif · 2025-11-27
> >
> > Thank you for addressing my concerns. I have changed my rating to 6.

---

### Official Review · Reviewer_TbE8 · 2025-10-28

**Soundness:** 3
**Presentation:** 2
**Contribution:** 2
**Rating:** 4
**Confidence:** 4

**Summary:**

This paper introduces S2AP, a plug-in method for adversarial pruning that minimizes sharpness in the score space. By perturbing and optimizing these scores to reduce loss landscape sharpness, S2AP stabilizes mask selection and enhances adversarial robustness.

**Strengths:**

1. The work extends the concept of sharpness minimization from the parameter space to the score-space setting.
2. It proposes a score-space sharpness minimization approach tailored for adversarial pruning.
3. The proposed method improves adversarial robustness and outperforms existing pruning baselines such as HARP and HYDRA.

**Weaknesses:**

1. The motivation is somewhat weak, and key concepts such as the sensitivity of top-k selection to score variations and the link between score-space sharpness and robustness are insufficiently explained.

2. The fairness of comparison may be questionable since it is unclear whether HARP and HYDRA are fine-tuned with AWP as in S2AP. Results under identical fine-tuning setups should be provided.

Minor：
3. The explanation of “trade-off in generalization” and its connection with ∆ is unclear.

4. The proposed method appears sensitive to hyperparameters (γ), which vary significantly across settings; this should be discussed as a limitation.

5. Incorrect citation formatting on page 1, line 43.

**Questions:**

The statement “minor score variations can lead to large changes in the selected top-k parameters” is not clearly explained. Could the authors provide a more intuitive explanation or a simple illustrative example to clarify this phenomenon?
Moreover, what is the relationship between the score-space loss landscape and adversarial robustness? Understanding this connection is essential to justify the motivation of the proposed score-space sharpness minimization.

Are HARP and HYDRA also fine-tuned using the AWP strategy? If not, the comparison in Table 1 and Table 2 might be unfair, since fine-tuning with AWP can significantly influence robustness. Could the authors report results where S2AP adopts the same fine-tuning setup as HARP and HYDRA to ensure a fair comparison?

On page 6, line 319, the authors state that “∆ remains mainly positive, showing that S2AP improves over Orig. without introducing a significant trade-off in generalization.” However, it is unclear what exactly the “trade-off in generalization” refers to. What is the relationship between ∆ and generalization?

---

> ### Author Response · Authors · 2025-11-21
> **Response by authors**
>
> We thank the reviewer for their helpful feedback. We respond to the listed weaknesses and questions below.
>
> **W1 & Q1 - Motivation behind S2AP:** We thank the reviewer for highlighting the need for a clearer explanation. We address both W1 and Q1 below.
> - *Why top-k selection is highly sensitive to small score variations:* During the pruning stage, score-based adversarial pruning methods optimize continuous importance scores $\mathbf s$, but the final subnet is determined by a discrete `top-k` operator that maps scores to a binary mask $M(\mathbf s,k)$. This introduces an inherent discontinuity: the pruning mask changes whenever two scores around the `k-th` threshold swap their order. As a result, even very small score variations can lead to discrete and potentially large structural changes in the selected pruning mask. We also quantify this phenomenon in Figures 1(b) and 4: the hamming distance computed on the masks across iterations suggests that standard AP methods, such as HARP and HYDRA, suffer from higher volatility (larger mask changes). In contrast, S2AP significantly smooths this volatility, providing empirical evidence that minimizing score-space sharpness, and thus reducing score sensitivity, stabilizes the `top-k` selection.
> - *Link between score-space loss landscape and adversarial robustness:* The relationship, in this case, stems from the fact that we are minimizing a robust loss objective $\hat{\mathcal{L}}$. Hence, by minimizing sharpness in the score-space loss landscape, we explore a region in score space where the adversarial robust loss remains low, even if the scores (and, as discussed above, the mask) are slightly perturbed.
>
> **W2 & Q2 - Fairness of the comparison:** We agree that it is critical to distinguish whether the improvement comes from the mask selection (the core contribution of S2AP) or merely from the fine-tuning strategy (AWP). To explicitly address this, we specify that we designed an ablation study already presented in Section 4.2 and Table 5, which compares the models immediately after the pruning stage and before any fine-tuning is applied. As a result, we find that the masks found by S2AP consistently yield higher robust accuracy than those selected by the original HARP and HYDRA methods across all sparsity levels and architectures. While we utilize AWP for fine-tuning in Tables 1 and 2 to align the two stages of the pipeline with the sharpness minimization philosophy, we believe Table 5 provides a fair and controlled comparison.
>
> **Q3: Clarification on the metric:** We thank the reviewer for asking to clarify. The $\Delta$ quantity is used in our analysis as a marker of whether improving robustness comes at the cost of noticeably degrading clean accuracy. In the adversarial robustness literature, it is common for robustness-oriented methods to introduce a trade-off between clean and robust accuracy, meaning that gains in adversarial robustness are obtained at the expense of significantly lower clean accuracy. We track that with $\Delta=(acc_{Orig.}-robust_{Orig.})-(acc_{S2AP}-robust_{S2AP})$. Hence, a positive $\Delta$ implies that S2AP’s gap is smaller than Orig.’s gap. We consistently find this quantity to be positive.
>
> **Final Remarks:** We thank the reviewer for the constructive feedback. We believe that the motivation and fairness concerns have been largely clarified, respectively, by our response and ablations, which confirm that S2AP’s gains are independent of finetuning.
> We hope this helps the reviewer reconsider their evaluation.

---

### Official Review · Reviewer_nDxS · 2025-10-31

**Soundness:** 2
**Presentation:** 3
**Contribution:** 2
**Rating:** 4
**Confidence:** 4

**Summary:**

This paper proposes a new Adversarial Pruning (AP) method, i.e., S2AP (Score-space Sharpness-aware Adversarial Pruning). The method explicitly minimizes “sharpness” in the score space, improving the stability of mask selection during pruning and thereby enhancing adversarial robustness. The authors extend the traditional concept of sharpness minimization in the parameter space to the optimization of space score, introducing the concept of “score-space sharpness minimization.”

**Strengths:**

1.	The idea of transferring “sharpness minimization” from the parameter space to the score space for mask optimization is novel.
2.	S2AP is a plug-in module that can be integrated into any score-based pruning framework without modifying the original objective.
3.	Extensive experiments demonstrate S2AP’s generality and robustness across different scenarios.
4.	They propose a “mask stability” metric to quantitatively verify that S2AP makes the mask search process smoother and more stable.

**Weaknesses:**

1.	The paper contains several formatting problems, including incorrect citation format (e.g, line 43) and missing punctuation (e.g. line 245).
2.	Limited comparison with existing sharpness-aware methods. While the authors acknowledge that S2AP is inspired by sharpness-aware approaches such as SAM [1] and AWP [2], no experiments compare S2AP directly with SAM or AWP under the same conditions. As a result, it remains unclear how much the proposed method truly improves adversarial robustness compared to these prior approaches.
3.	For S2AP’s finetuning, the authors use AWP approach to minimize sharpness in the classical weight space. While an ablation suggests that S2AP’s robustness improvement is not entirely due to AWP, part of the gain still comes from AWP approach, and the paper does not propose any novel contribution for the finetuning procedure itself.

[1] Foret et al., Sharpness-aware minimization for efficiently improving generalization. In International Conference on Learning Representations.

[2] Wu et al., Adversarial weight perturbation helps robust generalization. Advances in neural information processing systems.

**Questions:**

1.	S2AP is applied after 5 warm-up epochs. What is done during the 5 warm-up epochs and why exactly 5 epochs were chosen?
2.	Could the authors provide more experimental results comparing S2AP with SAM and AWP under the same conditions to clarify how much the proposed method truly improves adversarial robustness compared to these prior approaches？
3.	In Table 5, the authors report mask robust accuracy on CIFAR-10 and SVHN across sparsity levels using ResNet18, VGG-16, and WideResNet-28-4. However, clean accuracy under the same settings is not shown. Could the authors provide the corresponding clean accuracy results for these models and sparsity levels to give a more complete evaluation?

---

> ### Author Response · Authors · 2025-11-21
> **Response by authors**
>
> We appreciate the reviewer for noting the relevance and novelty of our work and for providing their feedback. We respond to the listed weaknesses and questions below.
>
> **W1 - Formatting:** We thank the reviewer for pointing out the formatting and punctuation issues. We will correct the citation on line 43 and the punctuation on line 245, and perform a thorough review to ensure consistent formatting.
>
> **W2 & Q2 - S2AP against AWP:** We respectfully note that we do compare S2AP against sharpness-aware methods under the same conditions. In particular, Section B.5 and Table 12 report experiments where we apply AWP-style sharpness minimization directly in weight space during pruning, using the same architectures, datasets, and sparsity levels as in our main S2AP experiments.
> These results demonstrate that minimizing weight-space sharpness alone does not match the improvements achieved by S2AP, and that explicitly minimizing score-space sharpness yields higher robustness gains and improved mask stability beyond what AWP provides.
>
> **W3 - Novelty in pruning and novelty in finetuning:** We agree that the novelty of our work does not lie in the finetuning procedure itself. Our contribution lies in introducing score-space sharpness minimization for pruning, rather than redefining the classical weight-space sharpness methods already established in the literature. We use AWP to maintain consistency with the finetuning stage and to align it with the pruning objective. Thus, while finetuning follows standard practice, the novelty resides in how sharpness is incorporated into the pruning phase.
>
> **Q1 - Warm-up:** During the 5 warm-up epochs, we train the model using standard adversarial training with the TRADES loss, following the common practice adopted in adversarial pruning works, such as HARP and HYDRA. This warm-up phase is necessary to obtain robust importance scores in the early pruning stages. Without it, early scores are too noisy, and applying S2AP directly leads to unstable mask selection and degraded performance. This is a standard choice also in weight-space sharpness minimization, where networks are primed with standard training before applying an AWP or SAM-like objective.
> We chose 5 epochs based on preliminary experiments, which showed that this duration is sufficient to stabilize the score distribution before the S2AP min–max update, while keeping the overall training cost low. Longer warm-ups did not yield measurable benefits, so we adopted 5 epochs as the minimal effective choice
>
> **Q2**: Answer in W2 response.
>
> **Q3 - Additional Table 5 results:** We thank the reviewer for pointing this out, and agree on the importance of showing the complete results for Table 5. We provide an overview here for CIFAR10 and update the main paper accordingly:
>
> ResNet18:
>
> | Sparsity (%) | HARP Orig.           | HARP-S2AP          | HYDRA Orig.      | HYDRA S2AP       |
> | ------------ | --------------- | --------------- | ----------- | --------------- |
> | **80**       | 83.19/48.88     | **82.03/49.55** | 82.13/48.56 | **82.87/48.98** |
> | **90**       | 82.98/49.42     | **83.12/49.60** | 80.55/47.41 | **82.26/48.06** |
> | **95**       | **82.26/49.04** | 82.48/48.43     | 78.98/45.55 | **79.47/45.61** |
> | **99**       | 72.97/40.99     | **75.46/41.86** | 69.63/35.15 | **69.66/36.74** |
>
> VGG16:
> | Sparsity (%) | HARP Orig.           | HARP-S2AP          | HYDRA Orig.      | HYDRA S2AP       |
> | ------------ | --------------- | --------------- | ----------- | --------------- |
> | **80**       | 75.78/41.93     | **76.51/42.84** | 74.86/40.31 | **76.15/41.39** |
> | **90**       | 73.89/41.69     | **75.86/42.11** | 73.19/38.12 | **75.17/40.61** |
> | **95**       | **73.55/40.21** | 74.68/39.13     | 62.30/31.81 | **72.86/38.03** |
> | **99**       | 52.59/24.22     | **72.76/36.41** | 40.77/20.54 | **60.91/29.67** |
>
> WRN-28-4
> | Sparsity (%) | HARP Orig.           | HARP-S2AP          | HYDRA Orig.      | HYDRA S2AP       |
> | ------------ | ----------- | --------------- | --------------- | --------------- |
> | **80**       | 82.97/50.45 | **83.24/50.59** | 82.59/50.31     | **83.05/50.41** |
> | **90**       | 81.82/50.56 | **82.66/50.79** | 80.71/47.75     | **81.92/49.30** |
> | **95**       | 80.44/49.07 | **80.82/49.37** | **79.82/46.97**     | 80.19/46.85 |
> | **99**       | 71.57/38.89 | **71.64/39.89** | 70.33/34.57 | 71.40/36.30     |
>
>
> **Final Remarks:** We thank the reviewer again for the constructive feedback. We believe the clarifications provided above address the concerns regarding motivation, comparisons with sharpness-aware methods, and finetuning. We hope these clarifications help consolidate the novelty and contribution of S2AP.

---

### Official Review · Reviewer_Bop5 · 2025-11-12

**Soundness:** 2
**Presentation:** 2
**Contribution:** 2
**Rating:** 0
**Confidence:** 4

**Summary:**

S2AP is a new plug-in method presented in this paper, for adversarial pruning that aims to improve robustness by minimizing score-space sharpness during mask selection. The authors argue that existing adversarial pruning methods suffer from unstable mask selection due to sharp local minima in the score-space loss landscape. S2AP introduces a min–max optimization over perturbed importance scores to flatten the loss surface and stabilize pruning decisions. The method is evaluated across multiple datasets (CIFAR-10, SVHN, ImageNet), architectures (ResNet, VGG, ViT), and pruning methods (HARP, HYDRA).

**Strengths:**

1. The paper identifies a clear and important issue, mask instability due to sharp score-space transitions. It provides empirical evidence of its impact on robustness.
2. Experiments span multiple datasets, models, and pruning methods, with consistent improvements in robust accuracy and mask stability.
3. S2AP is designed to integrate with existing pruning pipelines, making it practically useful.

**Weaknesses:**

My concerns are listed in following aspects.

1. The overall idea appears to be incremental, which is a direct extension of sharpness-aware minimization in weight-space. Applying this to score-space is intuitive but not conceptually novel. Similar ideas have already been explored in AdaSAP and S2-SAM, which also target sharpness minimization for sparse training.

2. The min–max formulation is plausible but lacks rigorous analysis. There is no convergence proof, no bounds on sharpness reduction, and no formal comparison to weight-space perturbation methods like AWP. The sharpness metric used (Hessian eigenvalues) is empirical and not tied to generalization guarantees.

3. The use of Hamming distance between masks across runs is a coarse proxy for stability. It does not capture functional similarity or robustness of the resulting subnetworks. More nuanced metrics (e.g., gradient similarity, spectral gap analysis would provide deeper insight.

4. Although ViT is included, S2AP is only applied to HYDRA due to HARP’s layer-wise constraints. This limits generalizability. Recent works on structured pruning for transformers show that fairness and robustness trade-offs are complex and require more tailored approaches.

5. The paper reports a ~15% increase in pruning time but does not analyze scalability or runtime implications for large models. This is especially relevant for transformer-based architectures and real-world deployment.

**Questions:**

Please refer to my weakness discussion for questions.

**Details Of Ethics Concerns:**

Public datasets.

---

> ### Author Response · Authors · 2025-11-19
> **Response by authors**
>
> We thank the reviewer for their feedback and address each of the weaknesses below.
>
> **W1 - S2AP novelty goes beyond sharpness minimization:** Our work is inspired by the general idea of sharpness minimization, but S2AP is indeed conceptually novel. Let us note that the referenced AdaSAP and S2-SAM approaches operate only in weight space and aim to learn a flatter weight minimum for training. In score-based adversarial pruning, however, weights are frozen, and the optimization is performed on importance scores.
> This difference is substantial: score-space defines a distinct, discontinuous landscape where small perturbations can completely change the selected mask. No prior sharpness-aware method addresses this setting, nor the resulting mask-selection instability, which is unique to score-based adversarial pruning. S2AP is the first to formulate and minimize score-space sharpness, and to show its direct effect on stabilizing mask search and improving robustness.
> Thus, while related in spirit, S2AP introduces a novel sharpness-aware mechanism tailored to the mask-selection problem, which is not covered by AdaSAP or S²-SAM.
>
> **W2 - Formulation and Hessian measure:** We acknowledge that our min–max formulation inherits the structure of AWP, and for this reason, we did not attempt to re-derive convergence guarantees already established in the literature. Our goal is to extend the adversarial perturbation principle to score space, which is a different optimization domain but follows the same single-step approximation used in AWP.
> Regarding sharpness metrics, we deliberately avoid relying solely on the Hessian eigenvalue. As we already specified in Section 4.3, this metric is used only for empirical evaluation, and its known limitations (described in detail in Section C.2) motivate our inclusion of the loss-difference metric from Andriushchenko et al. (ICML 2023).
>
> **W3 - Hamming is a well-established measure in pruning literature:** Our goal is to study the stability of mask selection, rather than the functional similarity between trained subnetworks. For this purpose, the Hamming distance is the standard metric used in prior pruning literature, as seen in prior work such as You et al. (ICLR 2020 spotlight) and Zhang et al. (ICML 2021). This is a crucial point, since the measure directly quantifies the variability of the selected parameters, which is precisely the object of study in score-based adversarial pruning.
> While more sophisticated stability metrics are an interesting direction for future work, the chosen measures are appropriate for evaluating mask-search dynamics, which is the central focus of our method.
>
> **W4 -  Layer-wise constraints are not a limitation of S2AP:** We would like to clarify that the decision to apply S2AP to HYDRA on ViTs is not a limitation of S2AP itself, but a consequence of HARP’s design, which imposes layer-wise sparsity schedules that are not directly compatible with transformer architectures without re-engineering the method.
> Regarding the remark on fairness–robustness trade-offs, we respectfully note that this point is unrelated to our contribution. Our work focuses on score-space sharpness and mask-search stability, rather than fairness metrics, and the reviewer does not cite specific works that demonstrate our method would be affected by such trade-offs. We kindly suggest providing curated references when pointing out potential weaknesses, so that the discussion is kept as constructive as possible.
>
> **W5 - Runtime:** We acknowledge that S2AP introduces a modest (~15%) overhead due to the additional adversarial step on scores. We expect that to be similar for ViTs and will include it in our final version of the paper.
>
> **Final Remarks:** Given these clarifications, which we believe directly and comprehensively address each of the raised weaknesses, we hope the reviewer may reconsider whether the initial, strongly negative score fully reflects the paper’s contributions and the evidence provided.
>
> ### References
>
> You, Haoran, et al. “Drawing early-bird tickets: Towards more efficient training of deep networks.” International Conference on Learning Representations, Spotlight. ICLR, 2020.
>
> Zhang, Zhenyu, et al. "Efficient lottery ticket finding: Less data is more." International Conference on Machine Learning. ICML, 2021.
>
> Andriushchenko, Maksym, et al. "A Modern Look at the Relationship between Sharpness and Generalization." International Conference on Machine Learning. ICML, 2023.

---

### Author Response · Authors · 2025-12-01
**Comments to reviewers and chairs.**

We would like to thank the reviewers, ACs, SACs, and PCs for their efforts on our submission.
Over the past days, we have been preparing an updated version of the manuscript with the goal of incorporating all possible feedback. In light of the complexity of the current review cycle and the reassignment of area chairs, we have decided to submit the revision now and integrate the points discussed with reviewers so far.

In reviewing the feedback, we identified three types of concerns:
- (i) comments referring to analyses or discussions that were already present in the manuscript but not sufficiently highlighted;
- (ii) comments arising from differing interpretations of our main contribution; and
- (iii) valid points that helped us improve the clarity and quality of the work.

To address all three categories, we updated the manuscript by (i) making existing contributions more visible, (ii) improving the conceptual framing and contextualization of S2AP, and (iii) incorporating the reviewers’ valid suggestions.

Among the most important updates, we:
- strengthened the motivation behind S2AP and its distinction from weight-space sharpness minimization;
- gave more spotlight to the finetuning ablation study comparing score-space and weight-space sharpness minimization;
- added the clean-accuracy results corresponding to Table 5.

We hope these refinements help consolidate the contributions of S2AP and support a smooth evaluation of our work.

---

### Meta-Review · Area_Chair_9TJy · 2025-12-28

**Summary:**

Across the reviews, the main concerns centre on novelty, rigour and evaluation completeness, and the clarity/fairness of comparisons. In particular, one reviewer (Bop5) remains strongly negative, largely on the grounds that the “overall idea appears incremental” and that the method “lacks rigorous analysis.”

Having considered the full set of reviews and the rebuttal, I believe the authors have addressed most points effectively. However, the criticism regarding rigorous analysis remains insufficiently resolved. The authors respond that the method “inherits the structure of AWP” and therefore they do not re-derive convergence guarantees. This does not adequately address the reviewer’s underlying request.

In weight space, flat minima have been linked to generalisation theories, most notably PAC-Bayesian analyses, which can yield explicit generalisation and, in some settings, robustness guarantees. By contrast, S2AP operates in the space of importance scores that determine a discrete mask via a top-k operator. The manuscript does not yet provide a corresponding theoretical perspective for how guarantees might be developed in this setting, and largely treats the point at a high level. I believe a more detailed discussion here, e.g., articulating an appropriate notion of score-space complexity, stability, or margin around the mask-selection boundary, and clarifying how such quantities could connect to generalisation/robustness, would significantly strengthen the paper.

**Reviewer Concerns:**

**Addressed**

1. Motivation (TbE8 and ASif):
The authors gave a clear discontinuity argument (continuous scores → discrete top-k mask; small score swaps near threshold cause large mask changes) and tied sharpness minimization to stability and robust loss.

2. Experiments Fairness (nDxS,TbE8):
The authors pointed to the pruning-stage ablation (Table 5 / Section 4.2) to isolate improvements from mask selection before fine-tuning, and clarified why AWP is used in fine-tuning for consistency.

3. Compute cost concern (ASif, Bop5):
The rebuttal pointed to a cost table (Appendix A.4 / Table 7) and reiterated the overhead is modest (~15%), addressing ASif/W3 and Bop5/W5 in substance (even if not fully satisfying for “scalability to very large models”).

4. Related-work distinction (ASif):
The authors clarified that prior work like AdvRush measures sharpness in different variables (e.g., input/architecture) whereas S2AP is explicitly score-space.

5. Presentation fixes (nDxS, TbE8):
Formatting/citation issues acknowledged and promised to fix.

6. Comparisons with AWP (nDxS)
authors argue Appendix/Table 12 compares weight-space sharpness during pruning (AWP-style) and shows S2AP is better.



several reviewers were already near threshold (rating 4 “marginally below but wouldn’t mind accept”), and at least one reviewer explicitly upgraded to 6 after the rebuttal. The remaining risk is that one reviewer (Bop5) remains highly negative largely due to rigor/novelty expectations and missing direct SAM/AWP comparisons, plus secondary concerns about metrics and scalability.



**Still outstanding**

1. convergence/analysis/bounds (Bop5):
The authors respond that the method “inherits the structure of AWP” and therefore they do not re-derive convergence guarantees. This does not adequately address the reviewer’s underlying request. In weight space, flat minima have been linked to generalisation theories, most notably PAC-Bayesian analyses, which can yield explicit generalisation and, in some settings, robustness guarantees. By contrast, S2AP operates in the space of importance scores that determine a discrete mask via a top-k operator. The manuscript does not yet provide a corresponding theoretical perspective for how guarantees might be developed in this setting, and largely treats the point at a high level. I believe a more detailed discussion here, e.g., articulating an appropriate notion of score-space complexity, stability, or margin around the mask-selection boundary, and clarifying how such quantities could connect to generalisation/robustness, would significantly strengthen the paper.

2. Mask stability metric (Bop5):
Bop5’s request for more nuanced stability/functionality measures is not directly addressed (authors defend Hamming as standard).

**Reviewer Scores:**

ASif: Rating explicitly changed to 6.

nDxS, TbE8: Likely to improve to 6, as the authors addressed their main concerns regarding motivation, fairness of comparison, and evaluation completeness.

Bop5: Likely to increase to 2, since several core concerns, e.g., the lack of rigorous theoretical analysis and the absence of more stability metrics, were not fully addressed in the rebuttal.

---

### Decision · Program_Chairs · 2026-01-26

Reject